# CLINIC: Towards High-quality Graph Out-Of-Distribution Detection

**Yifan Wang** [1]  **Haodong Zhang** [2]  **Changhu Wang** [3]  **Tao Ren** [2]  **Dongjie Wang** [4]  **Wei Ju** [5]  **Chong Chen** [6]
**Xian-Sheng Hua** [7]  **Xiao Luo** [8]

## Abstract

This paper studies the problem of graph out-of-distribution (OOD) detection, which aims to identify anomaly graphs out of a graph dataset. Prior efforts usually focus on the utilization of topological structures with unsupervised graph learning to foster typical pattern recognition, which overlooks the semantic structure preserved in contextual affinity neighborhoods. Towards this end, we propose a novel approach termed Contextual Affinity Exploration with Twin Concordance (CLINIC) for graph OOD detection. The core of CLINIC is to explore and exploit the contextual affinity of the graph data samples for discriminative graph representations. In particular, our CLINIC first builds a contextual affinity graph to depict the semantic structure in the hidden space. More importantly, we introduce high-order affinity to enhance geometric understanding of the structure by utilizing a meta-graph neural network. To enhance representation discriminability with high robustness, we introduce twin concordance learning, which not only minimizes the difference of affinity distributions across different views, but also encourages the consistency between contextually affinitive neighbors. Finally, we introduce a compression strategy to expand the decision boundary for enhanced separation between out-of-distribution and in-distribution graphs. Extensive experimental results further demonstrate the superiority of our CLINIC across ten real datasets.

[1]School of Artificial Intelligence and Data Science, University of International Business and Economics, Beijing, China [2]Software College, Northeastern University, Shenyang, China [3]Fred Hutchinson Cancer Research Center, Seattle, USA [4]Electrical Engineering and Computer Science, University of Kansas, Lawrence, USA [5]Peking University, Beijing, China [6]Hesicare Technology Co. Ltd, Shanghai, China [7]Institute of AI for Engineering, Tongji University, Shanghai, China [8]Department of Statistics, University of Wisconsin–Madison, Madison, USA. Correspondence to: Haodong Zhang <hdzhang819@163.com>, Xiao Luo <xiao.luo@wisc.edu>.

*Proceedings of the $43^{rd}$ International Conference on Machine Learning*, Seoul, South Korea. PMLR 306, 2026. Copyright 2026 by the author(s).

## 1. Introduction

Graphs serve as an indispensable tool for handling complex relational data across diverse fields, including molecular graph (Hao et al., 2020), biological system (Borgwardt et al., 2005) and citation network (Wang et al., 2022), etc. Building on the powerful representation capabilities of neural models, graph neural networks (GNNs) have emerged as a leading paradigm for encoding complex graph data (Kipf & Welling, 2017; Veličković et al., 2018; Xu et al., 2018). By leveraging a neighborhood-based message-passing mechanism to iteratively refine node representations, followed by a readout function to aggregate these updates into a comprehensive graph-level representation, GNNs are well-equipped to capture complex relational dependencies within the data and excel in various applications (Wu et al., 2020).

However, a key challenge with mainstream GNNs lies in their reliance on the i.i.d. assumption, which posits that the testing data is sampled from the same in-distribution (ID) as the training set. In real-world scenarios, especially when labeled graph data is limited, this assumption often fails (Ju et al., 2024c). As a result, GNN models typically struggle with out-of-distribution (OOD) graph samples that were not seen during training. Ideally, for a trustworthy graph learning system developed in real world, it is insufficient to only fit ID samples, the system must also recognize unknown OOD inputs during deployment (Wu et al., 2022). This underscores the importance of graph OOD detection, which helps determine whether a graph is from the ID or OOD, allowing the model to take appropriate precautions.

Recently, there has been a variety of methods proposed for OOD detection, especially in images (Hendrycks & Gimpel, 2017; Sehwag et al., 2021) or text domain (Zhou et al., 2021). For graph-structured data, detecting OOD graph samples is inherently more difficult due to both the structural and property patterns that can vary across different domains. Several pioneering studies have begun to explore the graph OOD detection task (Ding et al., 2026). For example, GraphDE (Li et al., 2022) formulates a generative procedure to capture distribution shift for graph OOD detection. AAGOD (Guo et al., 2023), GOOD-D (Liu et al., 2023) and GOODAT (Wang et al., 2024) further concentrate on the scarcity of graph OOD samples and propose unsuper-

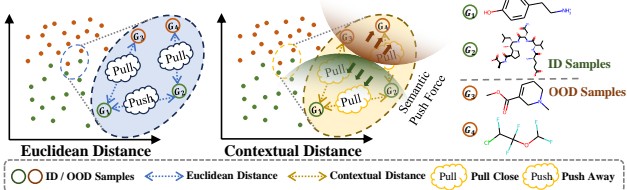

*Figure 1.* Comparison between the conventional Euclidean neighborhood and the contextually affinitive neighborhood. Under Euclidean distance metrics, cross-dataset samples with analogous feature patterns may exhibit proximity in the feature space. In contextually affinitive space, samples with diverse topological structures from the same dataset tend to show semantic similarities in a shared underlying distribution.

vised methods that are trained from scratch, post-hoc and at test-time respectively. HGOE (He et al., 2024) synthesizes outliers within ID graph samples to assist OOD detection training.

Despite the encouraging performance of these graph OOD detection methods, we argue they overlook the semantic structure preserved in contextual affinity neighborhoods. In fact, when two graph samples are drawn from the same data distribution, they are contextually affinitive, which means they may share similar relationships with a set of reference samples. As shown in Figure 1, $G_1$ and $G_2$ exhibit diverse topological structures (i.e., Benzene and Carbon Chain) results in their greater distance in Euclidean space. However, these samples belong to the same data distribution depicting contextual affinity and remain semantically distinct from samples in other datasets, despite they show similarities in topological structures (i.e., $G_1$ and $G_3$, $G_2$ and $G_4$). Therefore, there is a pressing need for an approach that could leverage the semantic structure preserved within neighborhood contexts for more effective graph OOD detection.

Towards this end, we propose a novel approach termed Contextual Affinity Exploration with Twin Concordance (CLINIC) for graph OOD detection, which employs an efficient retrieval process to mine more informative neighbors and encourages neighborhood concordance to learn a discriminative ID sample boundary. Specifically, given the input graph, we define the reference graphs as the current batch and estimate the corresponding high-order affinity to excavate more semantically relevant neighbors in a contextually affinitive space. Then, we consider the graph and its augmentation as two views, and group the graph into semantically meaningful clusters. Based on this, neighborhood-based concordance learning is proposed to explore the preserved semantic structure, enabling the model to produce a consistent and discriminative boundary for different underlying semantics within ID graphs. Finally, to achieve a more refined separation between ID and OOD graph samples, we introduce a decision boundary compression strategy for ID graphs, and thus, any test graph sample located out of the semantic boundary can be detected as an OOD graph.

The main contributions of this paper can be summarized as follows: ❶ *New Perspective.* We propose a new perspective of leveraging contextual affinity to enhance graph OOD detection, which provides a clear semantic structure in the hidden space. ❷ *Novel Methodology.* Our CLINIC not only enhances the geometric understanding with contextual high-order affinity, but also leverages twin affinity-aware concordance learning for robust and discriminative graph representations. ❸ *State-of-the-art Performance.* We extensively evaluate our CLINIC on various benchmark datasets, and the results confirm its superior performance.

## 2. Problem Definition & Preliminaries

**Notations.** Let $G = (\mathcal{V}, \mathcal{E}, \mathbf{X})$ denote an attributed undirected graph, where $\mathcal{V}$ and $\mathcal{E} \subseteq \mathcal{V} \times \mathcal{V}$ are the set of nodes and edges, and we assume the nodes and edges numer are given as $|\mathcal{V}| = n$ and $|\mathcal{E}| = m$, respectively. The node attributes are organized in a feature matrix $\mathbf{X} \in \mathbb{R}^{n \times d'}$, in which the row vector $\mathbf{x}_v$ corresponds to the $d'$-dimensional feature representation of node $v$. The graph connectivity is encoded by an adjacency matrix $\mathbf{A} \in \{0, 1\}^{n \times n}$, where $\mathbf{A}_{uv} = 1$ indicates there exist an edge between nodes $u$ and $v$, i.e., $(u, v) \in \mathcal{E}$; otherwise, $\mathbf{A}_{uv} = 0$.

**Graph-level OOD Detection.** We consider graph-level OOD detection under the setting where only ID graphs are available for training. Specifically, the training set is denoted as $\mathcal{D}_{train} = \{G_i\}_{i=1}^{N}$, where each graph $G_i = \{\mathcal{V}_i, \mathcal{E}_i, \mathbf{X}_i\}$ is drawn from the in-distribution $\mathbb{P}^{in}$. At test time, we assume there is also an OOD set $\mathcal{D}_{test}^{out}$ where graphs are drawn from an OOD distribution $\mathbb{P}^{out}$ and the test set contains both ID and OOD graphs, namely, $\mathcal{D}_{test} = \mathcal{D}_{test}^{in} \cup \mathcal{D}_{test}^{out}$, where samples in $\mathcal{D}_{test}$ follow $\mathbb{P}^{in}$. The objective of graph-level OOD detection is to train a detection model $f(\cdot)$ on $\mathcal{D}_{train}$ and then predict the source distribution of a graph sample $G' \in \mathcal{D}_{test}$ (i.e., $\mathbb{P}^{in}$ or $\mathbb{P}^{out}$). In practice, $f(\cdot)$ can be a scoring function, which can be defined as:

$$\text{Detection label} = \begin{cases} 1 \text{ (OOD)}, & \text{if } f(G') \geqslant \eta \\ 0 \text{ (ID)}, & \text{if } f(G') < \eta \end{cases} \quad (1)$$

where $\eta$ denotes the threshold. It should be noted that each graph from both $\mathbb{P}^{in}$ and $\mathbb{P}^{out}$ may fall into multiple categories, adding complexity to the detection task.

**Graph Neural Networks.** GNNs have recently become a prominent approach for learning from graph-structured data. Typically, GNNs operate through a series of message-passing layers, where each layer refines the node representations by aggregating and combining features from neighboring nodes. The process can be:

$$\mathbf{h}_v^{(l)} = \mathcal{M}^{(l)}(\mathbf{h}_v^{(l-1)}, \mathcal{A}^{(l)}(\{\mathbf{h}_u^{(l-1)}\}_{u \in \mathcal{N}(v)})), \quad (2)$$

where $\mathbf{h}_v^{(l)}$ denotes the refined embedding of node $v$ at $l$-th layer and we initialize $\mathbf{h}_v^0$ as $\mathbf{x}_v$ in practice. $\mathcal{N}(v)$ here

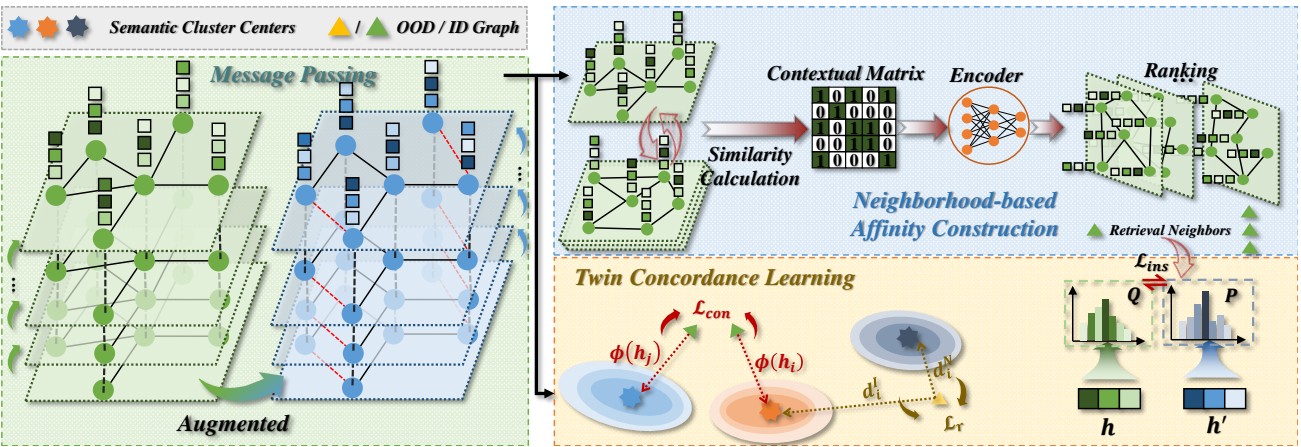

*Figure 2.* Overview of the proposed CLINIC. Our CLINIC consists of three modules: message passing, neighborhood-based affinity construction and contextual-aware clustering. During the message passing phase, features are extracted from both the original and augmented samples. These features are then fed into the neighbor-based affinity construction phase, where they are transformed into a high-order affinity space. Within this space, we retrieve the high-order affinitive neighbors for each sample and perform perturbation-aware concordance learning. Subsequently, the features undergo context-aware clustering via contextual-aware concordance learning. Finally, based on the clustering results, we execute decision boundary compression to yield the refined decision boundaries.

denote the neighbor set of $v$. $\mathcal{A}(\cdot)$ and $\mathcal{M}(\cdot)$ are the two basic functions that aggregate and combine information from neighbors at the previous layer. After $L$ layers of message-passing, a readout function can be applied to produce a graph-level representation $\mathbf{h}_G$, defined as:

$$\mathbf{h}_G = \text{READOUT}(\{\mathbf{h}_v^{(L)}\}_{v \in \mathcal{V}}). \tag{3}$$

Note that here READOUT$(\cdot)$ can be implemented via different types of pooling functions (i.e., mean and sum) in a permutation-invariant manner.

## 3. The Proposed CLINIC

### 3.1. Framework Overview

The basic idea of our CLINIC is to further consider the contextual relations among the graph data for OOD detection. As shown in Figure 2, our framework comprises three modules. Given the input graph, we retrieve the nearest neighbors by considering contextual relations among the whole batch data. Then, based on the transformed contextually affinitive space, we treat the input graph and its augmentation as two views and group the graph into semantically meaningful clusters. Neighborhood-based consistency is encouraged to achieve both instance-aware and contextual-aware concordance. This ensures the model produces a consistent and discriminative boundary for different underlying semantics within ID graphs. Finally, we progressively compress the decision boundary for effective detection.

### 3.2. Contextual Affinity Exploration for Semantic Structure Understanding

To capture the contextual relations of the graph, we transform the graph data points into a contextually affinitive

space (Yu et al., 2023; Zhang et al., 2024) and redefine the graph similarity to retrieve the nearest neighbors. Formally, given the input graph batch $\mathcal{B} = [G_1, \ldots, G_{|\mathcal{B}|}]$, we leverage the GNN-based encoder to capture the graph representation $\mathcal{H} = [\mathbf{h}_1, \ldots, \mathbf{h}_{|\mathcal{B}|}]$. Then, we construct the nearest neighbor enhanced affinity graph to explore the high-order contextual information between graphs, assuming that the more high-order contextual information two graphs share, the more similar the corresponding graph data points are in the transformed space.

**Neighborhood-based Affinity Construction.** For each graph in the batch, we retrieve its nearest neighbors using the dot product distance metric, namely $\langle \mathcal{H}, \mathcal{H}^{\mathrm{T}} \rangle$. In this way, the online reciprocal contextual matrix $\mathbf{R} \in \mathbb{R}^{|\mathcal{B}| \times |\mathcal{B}|}$ is estimated on the resulting $k$-NN graph, defined as:

$$\mathbf{R}(i,j) = \begin{cases} 1, \text{if } \mathbf{h}_i \in \mathcal{N}(\mathbf{h}_j, k_1) \wedge \mathbf{h}_j \in \mathcal{N}(\mathbf{h}_i, k_1) \\ 0, \text{if } \mathbf{h}_i \notin \mathcal{N}(\mathbf{h}_j, k_1) \wedge \mathbf{h}_j \notin \mathcal{N}(\mathbf{h}_i, k_1) \\ 0.5, \text{otherwise} \end{cases}$$
$$\tag{4}$$

where $\mathbf{h}_j \in \mathcal{N}(h_i, k_1)$ denotes $\mathbf{h}_j$ belongs to the $k_1$ nearest neighbors of $\mathbf{h}_i$ within $\mathcal{B}$. Based on the contextual matrix, we can transform the encoded graph representation into a contextually refined space, and the online global relation graph $G_\mathcal{B} = \{\mathcal{B}, \mathcal{E}_\mathcal{B}, \mathbf{X}_\mathcal{B}\}$ can be formulated as:

$$\mathcal{E}_\mathcal{B} = \{w_{ij} | \mathbf{h}_j \in \mathcal{N}(\mathbf{h}_i, k_2), i \in \{1, \ldots, |\mathcal{B}|\}\},$$
$$\mathbf{X}_\mathcal{B} = \{x_i | x_i = [\mathbf{R}_{i1}, \ldots, \mathbf{R}_{i|\mathcal{B}|}], i \in \{1, \ldots, |\mathcal{B}|\}\} \tag{5}$$

where $w_{ij} = \langle \mathbf{h}_i, \mathbf{h}_j \rangle$ denotes the edge weight and $\mathbf{h}_j \in \mathcal{N}(\mathbf{h}_i, k_2)$ indicates $\mathbf{h}_j$ is among the $k_2$ nearest neighbors of $\mathbf{h}_i$ in $\mathcal{B}$. In practice, we set $k_2$ far less than $k_1$.

**High-order Affinity Enhancement.** After the construction

of the global contextual affinity graph, we leverage a meta-GNN with the message-passing mechanism to further obtain the high-order contextual relations among the graphs, which can be computed as:

$$\mathbf{z}_i^{(l)} = \mathcal{M}^{(l)}(\mathbf{z}_i^{(l-1)}, \mathcal{A}^{(l)}(\{w_{ij}^\alpha \mathbf{z}_j^{(l-1)} | w_{ij} \in \mathcal{E}_\mathcal{B}\})), \quad (6)$$

where $w_{ij} \in \mathcal{E}_\mathcal{B}$ ensures that the nearest $k_2$ confident neighbors are selected to explore high-order relations, $\alpha$ is the hyper-parameter for weighted query expansion (Rade-nović et al., 2018; Yu et al., 2023). We initialize the embedding of $G_i$ within the contextually affinitive space as $\mathbf{z}_i^{(0)} = \mathbf{x}_i \in \mathbf{X}_\mathcal{B}$. After $L$ layers of propagation, the refined contextual feature can be $\mathbf{z}_i^L$, which explicitly contains the high-order affinitive information between graphs. Therefore, the affinitive neighborhood can be:

$$\mathcal{N}^\alpha(i, k) = \{G_j | \mathbf{z}_j^{(L)} \in \mathcal{N}(\mathbf{z}_i^{(L)}, k)\}. \quad (7)$$

This process ensures more informative $k$ reciprocal neighbors in contextually affinitive space.

### 3.3. Twin Concordance Learning for Representation Learning

Having acquired contextually affinitive neighbors, we aim to establish the decision boundary via self-supervision for graph OOD detection. Although contrastive learning has shown the capacity of capturing common patterns of ID graphs (Liu et al., 2023), we argue that most of the previous methods treat all other ID graphs as negative samples equally. This is contradictory to the fact that some so-called "negative" graphs can be similar or even in the same semantic class as the input graph. To tackle this issue, we compute the relational similarity distribution between the input graph and its contextually affinitive neighbors, and propose a perturbation- and contextual-aware concordance approach (Wei et al., 2021; Ju et al., 2023) to guide the model in learning consistent semantics for ID graph data.

**Perturbation-aware Concordance.** Perturbation-aware concordance encourages the input graph from different views to maintain the same relation distribution with the contextually affinitive neighbors. Specifically, given the input graph $G_i$, we randomly employ one of four fundamental graph data augmentation strategies (You et al., 2020), namely 1) *Dropping Node* 2) *Perturbation Edge* 3) *Masking Attribute* and 4) *Subgraph*, to generate another view of the graph $G_i'$ while preserving their intrinsic structural and attribute information. Then, we encode the original graph and its augmented view by the GNN-based encoder to get $\mathbf{h}_i$ and $\mathbf{h}_i'$, respectively. Here we encourage the two views to preserve consistent relational distributions for semantic alignment. Therefore, we denote the similarity distribution between the original graph $G_i$ and its contextually affinitive

neighbors $G_j \in \mathcal{N}^\alpha(i, k)$ as:

$$Q(i) = \frac{\exp\left(\langle \mathbf{h}_i, \mathbf{h}_j \rangle / \tau\right)}{\sum_{G_j \in \mathcal{N}^\alpha(i,k)} \exp\left(\langle \mathbf{h}_i, \mathbf{h}_j \rangle / \tau\right)}, \quad (8)$$

where $\mathbf{h}_j$ denotes the neighbor embedding and $\tau$ is a temperature hyper-parameter. And $Q(i)$ can be seen as the probability that $G_i$ selects $G_j$ as its matching context from affinitive neighbors. Similarly, we can also calculate the similarity distribution between the augmented view $G_i'$ and affinitive neighbors $G_j \in \mathcal{N}^\alpha(i, k)$:

$$P(i) = \frac{\exp\left(\langle \mathbf{h}_i', \mathbf{h}_j \rangle / \tau\right)}{\sum_{G_j \in \mathcal{N}^\alpha(i,k)} \exp\left(\langle \mathbf{h}_i', \mathbf{h}_j \rangle / \tau\right)}. \quad (9)$$

For each graph $G_i$, we impose the consistency between the probability distribution $P$ and $Q$ (Wei et al., 2021; Ju et al., 2023), and the loss function can be calculated as:

$$\mathcal{L}_{ins} = \frac{1}{2N} \sum_{G_i} (D_{KL}(P(i)\|Q(i)) + D_{KL}(Q(i)\|P(i))). \quad (10)$$

**Contextual-aware Concordance.** Compared to maintaining graph data correlations, contextual-aware concordance regards the local neighborhood as a semantic group and pushes the model to learn a consistent and discriminative boundary for the ID graph sample (Van Gansbeke et al., 2020; Huang et al., 2020). Toward this end, we introduce a clustering function $\phi$ that is parameterized by a neural network with softmax on the output to classify graph $G_i$ and its contextually affinitive neighbors $G_j \in \mathcal{N}^\alpha(i, k)$ together, namely defined as $\phi(\mathbf{h}_i) \in [0, 1]^C$, which performs a soft assignment over the clusters $\mathcal{C} = \{1, \ldots, C\}$. And the contextual-aware consistency loss can be defined as:

$$\mathcal{L}_{con} = -\frac{1}{N} \sum_{G_i} \sum_{G_j} \log\langle \phi(\mathbf{h}_i), \phi(\mathbf{h}_j) \rangle + \lambda \sum_{c \in \mathcal{C}} \phi_c' \log \phi_c', (11)$$

with

$$\phi_c' = \sum_{G_i} \phi_c(\mathbf{h}_i). \quad (12)$$

Note that the first term of loss imposes that each graph $G_i$ and its neighboring graph $\mathcal{N}^\alpha(i, k)$ make the same semantic group prediction. This term reaches its maximum when the predictions are one-hot (confident) and are assigned to the same cluster (consistent). Meanwhile, we introduce an entropy loss as the second term to avoid trivial solutions of $\phi$, which assigns all samples to a single cluster. Here $\phi_c(\mathbf{h}_i)$ denotes the probability of $G_i$ assigned to cluster $c$.

### 3.4. Decision Boundary Compression for Representation Enhancement

With the learned cluster structure of the ID graph data serving as the decision boundary, we focus on capturing holistic

semantics. This emphasizes that farther samples located at the cluster boundary still retain separability. Inspired by the silhouette score (Rousseeuw, 1987; Yu et al., 2023), which evaluates clustering quality by comparing the intra-cluster distance $d^I$ with the distance to the nearest neighboring cluster $d^N$ in an offline setting, we introduce an approximate online version to compress the decision boundary of the ID graph data:

$$d_i^I = \|\mathbf{h}_i - \boldsymbol{\mu}_{\phi(\mathbf{h}_i)}\|_2, \quad d_i^N = \|\mathbf{h}_i - \boldsymbol{\mu}_{\phi'(\mathbf{h}_i)}\|_2, \quad (13)$$

where

$$\phi'(\mathbf{h}_i) = \underset{c \in \{1,\dots,C\}\setminus\{\phi(\mathbf{h}_i)\}}{\arg\min} \|\mathbf{h}_i - \boldsymbol{\mu}_c\|, \quad (14)$$

with $\boldsymbol{\mu}_c$ denoting the center of cluster $c$ obtained from last epoch. The boundary compression loss can be defined as:

$$\mathcal{L}_r = \sum_{G_i} r_i, \; r_i = 1 - \frac{d_i^N - d_i^I}{\max(d_i^I, d_i^N)}, \quad (15)$$

where $r_i$ denotes the compression ratio and a larger $r_i$ indicates that the graph sample $G_i$ is more likely located at the boundary. Therefore, we minimize the compression ratio to impose the ID graph sample close to its cluster center.

### 3.5. Summarization

The overall objective is a combination of perturbation-aware consistency regularization loss, contextual-aware consistency regularization loss and decision boundary compression loss. Formally, the final loss of our CLINIC is:

$$\mathcal{L} = \mathcal{L}_{ins} + \beta \mathcal{L}_{con} + \gamma \mathcal{L}_r, \quad (16)$$

where $\beta$ and $\gamma$ denote the coefficients used to control their respective contributions. During the test phase, we directly employ boundary compression ratio $r$ as the OOD score of the test graph sample. The learning procedure of our proposed CLINIC is illustrated in Algorithm 1 in Appendix A.

### 3.6. Theoretical Analysis

We further present a theoretical analysis of our CLINIC. Let the parameter of the model be denoted as $\theta$. The compression ratio for the $i$-th graph corresponding to the optimal parameter $\theta^*$ is defined as $r_i(\theta^*)$. Suppose there are $C$ clusters in the training set. We consider the graph from the OOD in the test set as belonging to a new cluster. For the $i$-th graph in $\mathcal{D}_{\text{test}}^{\text{out}}$, let the graph representation be denoted by $\mathbf{h}_i = \mathbf{h}_i(\theta^*)$. With this notation, we define:

$$\boldsymbol{\mu}_{\text{out}} = \text{mean}\left\{\boldsymbol{h}_i(\theta^*) : G_i \in \mathcal{D}_{\text{test}}^{\text{out}}\right\}. \quad (17)$$

Next, we define the expected compression ratios for the ID and OOD graphs, respectively, as:

$$R_{\text{in}} = \mathbb{E}_{\text{in}}(r(\theta^*)), \quad R_{\text{out}} = \mathbb{E}_{\text{out}}(r(\theta^*)). \quad (18)$$

Our next goal is to show that the compression ratio for the OOD graphs is larger than that for the ID graphs, i.e., $R_{\text{out}} > R_{\text{in}}$. Thus, we can choose a threshold $\eta$ such that if $r(\theta^*) > \eta$, the graph is classified as OOD. The key term in $r(\theta^*)$ is $d_i^N(\theta^*) - d_i^I(\theta^*)$. For simplicity, we denote $d_i^N(\theta^*) - d_i^I(\theta^*)$ as $D$. We will demonstrate that the expected value of this term is smaller for the OOD graphs compared to the ID graphs, i.e., $D_{\text{out}} < D_{\text{in}}$. Specifically, we have:

$$D_\Delta = \mathbb{E}_\Delta\left(d_i^N(\theta^*) - d_i^I(\theta^*)\right), \Delta \in \{\text{in}, \text{out}\} \quad (19)$$

**Theorem 3.1.** *Assume that*

$$\mathbb{E}_{out}\left(\|\boldsymbol{\mu}_{out} - \boldsymbol{\mu}_{\phi'(h_i)}\|_2\right) - \mathbb{E}_{out}\left(\|\boldsymbol{\mu}_{out} - \boldsymbol{\mu}_{\phi(h_i)}\|_2\right) \quad (20)$$

*is sufficiently smaller than $\mathbb{E}_{in}\left(\|\boldsymbol{\mu}_{\phi(h_i)} - \boldsymbol{\mu}_{\phi'(h_i)}\|_2\right)$, where the expectation is taken at the optimal parameter $\theta^*$. Then, we have $D_{out} < D_{in}$.*

Theorem 3.1 implies that the $D_{\text{out}} < D_{\text{in}}$, suggesting that the compression ratio for the OOD graphs is larger than that for the ID graphs. Therefore, we can choose a threshold $\eta$ such that if $r > \eta$, the graph is classified as OOD. Note that the analysis in Theorem 3.1 is based on the optimal parameter $\theta^*$. In practice, we use the parameter $\hat{\theta}$ that minimizes the empirical loss.

**Theorem 3.2.** *Assume that*

$$\sum_{G_i} r_i(\hat{\theta}) - \sum_{G_i} r_i(\theta^*) \le 0. \quad (21)$$

*Then, we have*

$$\mathbb{E}_{in}\left(r(\hat{\theta})\right) - \mathbb{E}_{in}\left(r(\theta^*)\right) \le C\sqrt{\frac{d_{VC}}{n}}, \quad (22)$$

*where $C$ denotes a positive constant, $n$ is the number of training samples, and $d_{VC}$ is the Vapnik-Chervonenkis (VC) dimension of the neural network.*

Since the parameter $\hat{\theta}$ is obtained by minimizing the empirical loss, it is reasonable to assume that the empirical compression ratio at $\hat{\theta}$ is less than or equal to that at $\theta^*$. Under this assumption, Theorem 3.2 demonstrates that the expected compression ratio at $\hat{\theta}$ is close to that at $\theta^*$. Typically, the VC dimension of the neural network is related to the number of parameters (see Bartlett et al. (2021) for more discussion). Similarly, we have the results for the OOD graphs. The proof of this theorem based on empirical process theory is provided in the Appendix B.

*Table 1.* AUC-based evaluation of OOD detection methods, with results reported as mean ± std. The best and second-best performances are denoted by **bold** and underline, respectively.

| ID Dataset / OOD Dataset | BZR COX2 | PTC-MR MUTAG | AIDS DHFR | ENZYMES PROTEIN | IMDB-M IMDB-B | Tox21 SIDER | FreeSolv ToxCast | BBBP BACE | ClinTox LIPO | Esol MUV |
|---|---|---|---|---|---|---|---|---|---|---|
| PK-LOF | 42.22±8.39 | 51.04±6.04 | 50.15±3.29 | 50.47±2.87 | 48.03±2.53 | 51.33±1.81 | 49.16±3.70 | 53.10±2.07 | 50.00±2.17 | 50.82±1.48 |
| PK-OCSVM | 42.55±8.26 | 49.71±6.58 | 50.17±3.30 | 50.46±2.78 | 48.07±2.41 | 51.33±1.81 | 48.82±3.29 | 53.05±2.10 | 50.06±2.19 | 51.00±1.33 |
| PK-iF | 51.46±1.62 | 54.29±4.33 | 51.10±1.43 | 51.67±2.69 | 50.67±2.47 | 49.87±0.82 | 52.28±1.87 | 51.47±1.33 | 50.81±1.10 | 50.85±3.51 |
| WL-LOF | 48.99±6.20 | 53.31±8.98 | 50.77±2.87 | 52.66±2.47 | 52.28±4.34 | 51.92±1.58 | 51.47±4.23 | 52.80±1.91 | 51.29±3.40 | 51.26±1.31 |
| WL-OCSVM | 49.16±4.51 | 53.31±7.57 | 50.98±2.71 | 51.77±2.21 | 51.38±2.39 | 51.08±1.46 | 50.38±3.81 | 52.85±2.00 | 50.77±3.69 | 50.97±1.65 |
| WL-iF | 50.24±2.49 | 51.43±2.02 | 50.10±0.44 | 51.17±2.01 | 51.07±2.25 | 50.25±0.96 | 52.60±2.38 | 50.78±0.75 | 50.41±2.17 | 50.61±1.96 |
| InfoGraph-iF | 63.17±9.74 | 51.43±5.19 | 93.10±1.35 | 60.00±1.83 | 58.73±1.96 | 56.28±0.81 | 56.92±1.69 | 53.68±2.90 | 48.51±1.87 | 54.16±5.14 |
| InfoGraph-MD | 86.14±6.77 | 50.79±8.49 | 69.02±11.67 | 55.25±3.51 | 81.38±1.14 | 59.97±2.06 | 58.05±5.46 | 70.49±4.63 | 48.12±5.72 | 77.57±1.69 |
| GraphCL-iF | 60.00±3.81 | 50.86±4.30 | 92.90±1.21 | 61.33±2.27 | 59.67±1.65 | 56.81±0.97 | 55.55±2.71 | 59.41±3.58 | 47.84±0.92 | 62.12±4.01 |
| GraphCL-MD | 83.64±6.00 | 73.03±2.38 | 93.75±2.13 | 52.87±6.11 | 79.09±2.73 | 58.30±1.52 | 60.31±5.24 | 75.72±1.54 | 51.58±3.64 | 78.73±1.40 |
| OCGIN | 76.66±4.17 | 80.38±6.84 | 86.01±6.59 | 57.65±2.96 | 67.93±3.86 | 46.09±1.66 | 59.60±4.78 | 61.21±8.12 | 49.13±4.13 | 54.04±5.50 |
| GLocalKD | 75.75±5.99 | 70.63±3.54 | 93.67±1.24 | 57.18±2.03 | 78.25±4.35 | 66.28±0.98 | 64.82±3.31 | 73.15±1.26 | 55.71±3.81 | 86.83±2.35 |
| GOOD-D | 94.99±2.25 | 81.21±2.65 | 99.07±0.40 | 61.84±1.94 | 79.94±1.09 | 66.50±1.35 | 80.13±3.43 | 82.91±2.58 | 69.18±3.61 | 91.52±0.70 |
| HGOE | 95.00±2.70 | 82.06±1.63 | 99.28±0.34 | 64.44±2.19 | 81.74±2.25 | 68.24±0.60 | **82.89**±2.33 | 83.46±1.79 | 70.09±1.52 | 92.64±2.44 |
| CGOD | 95.47±3.85 | 81.42±2.04 | 98.17±0.28 | 61.52±1.62 | 79.02±3.80 | 69.10±1.58 | 81.72±1.07 | 81.75±1.43 | 65.13±2.61 | 89.68±3.02 |
| CLINIC | **96.32**±1.16 | **82.60**±2.14 | **99.42**±0.16 | **64.86**±2.48 | **82.44**±1.76 | **70.69**±0.42 | 82.00±1.16 | **84.21**±1.07 | **71.31**±2.10 | **93.11**±1.06 |

*Table 2.* AUC performance of CLINIC and its variants in the ablation study, reported in percent as mean ± std.

| $\mathcal{L}_{ins}$ | $\mathcal{L}_{con}$ | $\mathcal{L}_r$ | BZR COX2 | PTC-MR MUTAG | AIDS DHFR | ENZYMES PROTEIN | IMDB-M IMDB-B | Tox21 SIDER | FreeSolv ToxCast | BBBP BACE | ClinTox LIPO | Esol MUV |
|---|---|---|---|---|---|---|---|---|---|---|---|---|
| ✓ | ✗ | ✗ | 80.57±5.15 | 79.71±7.71 | 95.64±1.14 | 54.26±3.10 | 75.02±1.80 | 69.01±0.51 | 64.50±3.12 | 73.71±1.08 | 58.06±2.41 | 88.04±1.81 |
| ✗ | ✓ | ✗ | 77.41±6.38 | 76.42±5.88 | 97.10±2.03 | 53.27±1.67 | 74.08±2.44 | 68.65±0.59 | 67.54±5.61 | 76.42±2.12 | 58.22±2.90 | 88.28±1.43 |
| ✗ | ✗ | ✓ | 94.52±1.47 | 79.11±4.81 | 98.06±0.32 | 58.86±3.28 | 79.29±1.57 | 69.13±0.85 | 69.42±4.05 | 79.68±0.87 | 62.57±1.58 | 90.69±0.71 |
| ✓ | ✓ | ✗ | 80.96±5.88 | 75.98±6.34 | 97.01±1.93 | 59.57±3.25 | 79.67±3.29 | 69.40±0.78 | 72.38±4.74 | 81.56±1.17 | 69.35±6.23 | 91.25±1.39 |
| ✓ | ✗ | ✓ | 95.73±1.88 | 79.66±4.91 | 98.92±0.27 | 61.25±2.70 | 81.54±2.43 | 69.95±0.79 | 78.15±4.32 | 83.76±1.26 | 67.93±4.90 | 92.66±0.83 |
| ✗ | ✓ | ✓ | 95.78±1.43 | 82.48±2.56 | 98.90±0.39 | 61.76±5.82 | 80.79±3.13 | 69.78±0.55 | 76.37±4.07 | 65.80±1.18 | 70.20±1.01 | 92.53±1.42 |
| ✓ | ✓ | ✓ | **96.32**±1.16 | **82.60**±2.14 | **99.42**±0.16 | **64.86**±2.48 | **82.44**±1.76 | **70.69**±0.42 | **82.00**±1.16 | **84.21**±1.07 | **71.31**±2.10 | **93.11**±1.06 |

## 4. Experiments

### 4.1. Experiment Setup

**Datasets.** We construct ten dataset pairs in total, including five from OGB (Hu et al., 2020) and five from TU (Morris et al., 2020). The datasets within each pair are semantically related and feature-compatible, yet exhibit distribution shifts. For each pair, 90% of samples from one dataset are used as ID samples for training, while the remaining samples, along with an equivalent number of OOD samples from the other dataset, are used for testing. To evaluate performance in anomaly detection, we adopt 14 datasets selected from the TU dataset (Morris et al., 2020). Following the experimental protocol of prior studies (Ma et al., 2022; Liu et al., 2023), we construct the normal training set using samples from the majority class, and regard samples from the minority or pre-defined anomalous class as anomalies.

**Baselines.** We compare CLINIC against 15 baselines. Specifically, these approaches generally fall into three main groups: **non-deep two-step methods** (i.e., *Weisfeiler-Lehman kernel (WL)* (Shervashidze et al., 2011) and *propagation kernel (PK)* (Neumann et al., 2016) are used as feature extractors, while *one-class SVM (OCSVM)* (Manevitz & Yousef, 2001), *isolation forest (iF)* (Liu et al., 2008) and *local outlier factor (LOF)* (Breunig et al., 2000) are used as detectors), **deep two-step methods** (i.e., *GraphCL* (You et al., 2020) and *InfoGraph* (Sun et al., 2020) are used as feature extractors, while *iF* and *Mahalanobis distance-based*

*(MD)* are used as detectors), and **end-to-end methods** (i.e., *GLocalKD* (Ma et al., 2022), *OCGIN* (Zhao & Akoglu, 2023), *HGOE* (He et al., 2024), *GOOD-D* (Liu et al., 2023), and CGOD (Lin et al., 2025)).

**Evaluation and Implementation.** Following the evaluation protocol in previous work (Liu et al., 2023; He et al., 2024), we assess model performance using the area under the ROC curve (AUC). All experiments are conducted 5 times, and we report the mean AUC along with the standard deviation. More experimental details can be found in Appendix C. The code is available at `https://github.com/sodarin/CLINIC-for-OOD-detection`.

### 4.2. Performance Comparison

Table 1 presents the results of CLINIC against all baselines. Based on these results, we make the following observations. ❶ Compared to non-deep two-step methods, deep learning-based approaches consistently deliver better performance by leveraging high-dimensional features and topological information. ❷ End-to-end methods typically outperform two-step detection approaches, suggesting that integrating the feature extractor and detector enhances the refinement of tailored representations. ❸ Our CLINIC achieves superior performance over all competing baselines on most datasets. Notably, it achieves average AUC improvements of 1.16% and 2.65% relative to two baselines HGOE and GOOD-D, respectively. We attribute the improvements to three reasons. ❶ The utilization of contextual affinity en-

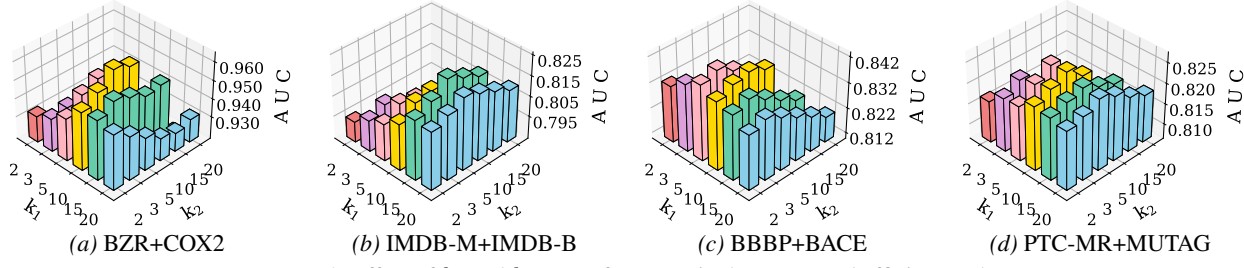

*Figure 3.* Effect of $k_1$ and $k_2$ on performance in the contextual affinity graph.

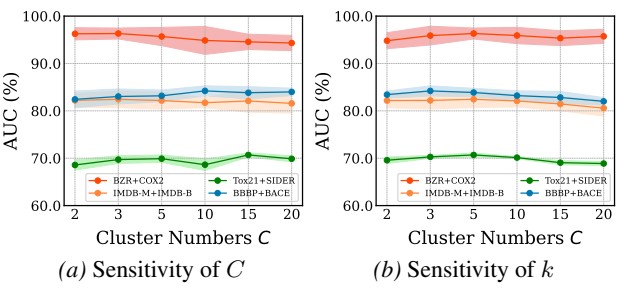

*Figure 4.* Effect of the number of clusters $C$ and affinitive neighbors $k$ on model performance.

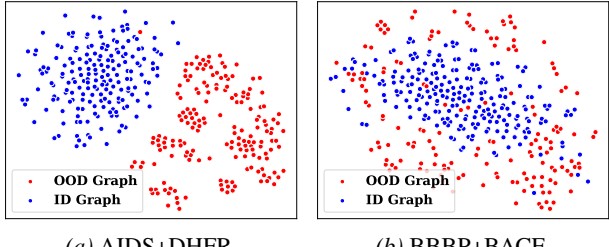

*Figure 5.* t-SNE visualization of graph representations produced by our model.

hances the understanding of the semantic structure within the ID dataset, fostering greater compactness in the underlying feature space. ❷ The introduction of twin affinity-aware concordance learning further reveals the preserved semantic structure, enabling robust and discriminative graph representations. ❸ The proposed decision boundary compression strategy contributes to refining implicit separation between OOD and ID samples. To provide a more comprehensive evaluation, we further conduct additional analysis of our model's anomaly detection performance in the Appendix D.

## 4.3. Ablation Study

In this section, we present ablation studies to illustrate the impact of the key components. We perform experiments on all combinations of loss functions. From the results shown in Table 2, we conclude that: ❶ Each component significantly contributes to overall performance, with decision boundary compression having the greatest impact due to its contribution in refining discriminative boundary. ❷ The combination of two components typically yields better results than simply using individual component. ❸ CLINIC with all components achieves the highest performance. These observations highlight the effectiveness of jointly utilizing the carefully designed components.

## 4.4. Parameter Analysis

**Analysis of contextual affinity graph construction.** We first analyze the optimal value of $k_1$ and $k_2$ in the construction of affinity graph in the range of $\{2, 3, 5, 10, 15, 20\}$ (note that $k_2 < k_1$). As shown in Figure 3, we have the following conclusions. ❶ $k_2$ that is too small or too close to $k_1$ can result in performance degradation as it either limits

information integration or the introduction of noise. ❷ The optimal values of both $k_1$ and $k_2$ vary across datasets. We conjecture that the best selections are closely influenced by the intrinsic properties of ID datasets, such as the class cardinality or typical patterns. Based on our experience, we recommend that $k_1 \in \{5, 10, 15\}$ and $k_2 \in \{3, 5\}$ usually yield promising performance.

**Analysis of cluster numbers.** We then study the sensitivity of CLINIC w.r.t. the cluster number $C$ in the range of $\{2, 3, 5, 10, 15, 20\}$. As shown in Figure 4(a), we discover that the optimal suitable choice of $C$ varies across dataset pairs, indicating its inherent correlations with the size of datasets. For example, the best result can be achieved with $C = 15$ in Tox21+SIDER, while a smaller $C$ (i.e., $C = 3$) is preferred by BZR+COX2. This variation may be related to the underlying semantic patterns of the datasets. Nevertheless, CLINIC is not sensitive to $C$.

**Analysis of reciprocal neighbor numbers.** We further investigate the sensitivity of the number of reciprocal neighbor $k$ in the range of $\{2, 3, 5, 10, 15, 20\}$. From Figure 4(b), we conclude that moderately incorporating affinitive neighbors enhances overall performance by effectively leveraging higher-order contextual information. However, excessive incorporation of reciprocal neighbors may introduce noise from false positive samples, resulting in a slight drop in performance. Overall, the performance remains relatively stable, with the best results typically observed when $k = 3, 5$.

## 4.5. Case Study

**Visualization on learned embeddings.** We use 2D t-distributed stochastic neighbor embedding (t-SNE) to visualize the learned embeddings of our model. As shown in

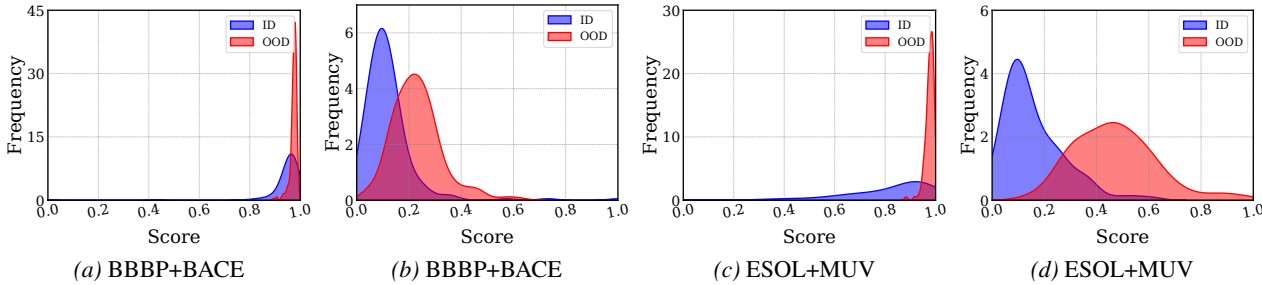

*Figure 6.* The score distributions before training (a, c) and after training (b, d) with our method.

Figure 5, on the one hand, the ID samples are well concentrated into a cluster with distinct boundary, exhibiting tight within-cluster aggregation. On the other hand, the OOD samples are positioned far from the cluster center of the ID samples, highlighting clearer distinctions. These phenomena provide explicit separation between ID and OOD samples, facilitating effective detection.

**Visualization on score distributions.** To intuitively evaluate the discriminative power of the model, we plotted the score distributions for the ID and OOD test samples. The visualization in Figure 6 reveals that our model clearly separates the peaks of the ID samples from the OOD samples with relatively less overlap area, which yields a promising boundary for distinguishing between the ID and the OOD samples. This observation further demonstrates the effectiveness of `CLINIC`.

## 5. Related Work

**Graph Neural Networks.** GNNs offer a promising approach for learning from graph-structured data and are gaining increasing attention in recent times (Hamilton et al., 2017; Veličković et al., 2018; Xu et al., 2018; Ju et al., 2024a). From the perspective of learning paradigm, GNNs can be categorized into two types: 1) Spectral-based GNNs and 2) Spatial-based GNNs. Spectral-based GNNs aim to learn graph convolution through spectral graph theory (Kipf & Welling, 2017; Wu et al., 2019). For example, GCN (Kipf & Welling, 2017) performs convolution using the graph Laplacian matrix to define filters that capture the graph structure. Spatial-based GNNs concentrate on aggregating and transforming local information (Veličković et al., 2018; Xu et al., 2018). For example, GIN (Xu et al., 2018) learns node representations by aggregating features from neighboring nodes in a manner that is sensitive to the graph structure, facilitating effective discrimination between different topologies. Despite the growing popularity of GNNs, several concerns have recently emerged. Here, we focus on OOD detection, aiming to provide a reliable learning framework for identifying OOD samples.

**Out-of-distribution Detection.** OOD detection involves identifying samples that significantly differ from the train-

ing distribution, helping to prevent model misclassifications and enhance robustness. Despite their success in images (Hendrycks & Gimpel, 2022) and text (Zhou et al., 2021) data, OOD detection in graph data still faces considerable challenges due to the complex topological structures that hinder effective pattern recognition. AAGOD first (Guo et al., 2023) proposed a parameterized amplifier to highlight prominent patterns, and thereby enhancing the distinction between OOD and ID data. GOOD-D (Liu et al., 2023) employs hierarchical contrastive learning with structural features exposed to ID samples during training. HGOE (He et al., 2024) integrates realistic graph data from external distribution while synthesizing internal outliers within ID data. However, previous efforts primarily focus on utilizing topological structure similarities to identify structural patterns for OOD detection, overlooking the underlying contextual affinities within the same data distribution, which results in suboptimal performance.

**Concordance Learning.** Concordance learning, which promotes consistent predictions on unlabeled data and its corresponding augmented versions, is a crucial component of semi-supervised learning approaches heavily used in image processing (Sajjadi et al., 2016; Wei et al., 2021; Zhang et al., 2023). Recently, motivated by its fascinating capability in capturing self-supervised information, concordance learning has also been widely introduced in graph data. For example, TGNN (Ju et al., 2022) leverages KL divergence to promote consistency between the similarity distributions of each unlabeled graph and the labeled graphs stored in the memory bank. HEAL (Ju et al., 2024b) utilizes concordance learning to mutually supervise and reinforce the line graph view and the hypergraph view. To the best of our knowledge, we are the first to apply concordance learning to the OOD task, aiming to explore the semantic structure within ID graphs in contextually affinitive space.

## 6. Conclusion

In this paper, we introduce a novel approach called `CLINIC` for graph OOD detection, focusing on semantic structure—an aspect often overlooked in previous research and equally important as topological structure. `CLINIC` first constructs a contextual affinity graph, which is processed

by a meta-graph neural network to reveal the underlying semantic structure through high-order affinity. Subsequently, we leverage twin concordance learning to foster robust and discriminative graph representations. Finally, we propose a compression strategy to achieve refined separation between OOD and ID samples. Our extensive experiments on various real-world datasets demonstrate that `CLINIC` yields significant performance in OOD detection tasks.

## Impact Statement

The proposed `CLINIC` improves graph out-of-distribution detection by leveraging contextual affinity and concordance learning, overcoming the limitations of neglecting semantic structure within the same data distribution. By exploring high-order semantically relevant neighbors in the contextually affinitive space and capturing consistency representations through concordance learning, `CLINIC` enhances the learning for robust and discriminative graph representations. This dual approach incorporates twin affinity-aware concordance learning into the excavation of semantic structures in contextually affinitive space for more insightful and accurate graph OOD detection. This work further shows promising potential for broad applications under open-world scenarios, such as anomaly detection and graph classification.

## Acknowledgements

Tao Ren is supported by the National Natural Science Foundation of China (62276058, 41774063), the Fundamental Research Funds for the Central Universities (N25GFZ011). Yifan Wang is supported by the Fundamental Research Funds for the Central Universities in UIBE (Grant No. 23QN02) and the Humanities and Social Sciences Research Fund of the Ministry of Education of China under Grant 25YJCZH275.

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

## A. Training Details

---

**Algorithm 1** The overall training process of `CLINIC`

---

**Input**: Training set $\mathcal{D}_{train} = \{G_1, G_2, \ldots, G_N\}$ where graph $G_i = \{\mathcal{V}_i, \mathcal{E}_i, X_i\}$ is sampled from ID dataset.

**Output**: OOD scores $r_i$ of each graph $G_i$ in test dataset $\mathcal{D}_{test} = \{G_1, G_2, \ldots, G_N\}$ sampled from ID and OOD dataset.

1: **while** not converged **do**
2:     Divide the training set into batches $\mathcal{B}$ randomly; form two graph views from the original graphs and the augmented graphs denoted as $\{G_1, G_2, \ldots, G_{\mathcal{B}}\}$ and $\{G_1^{'}, G_2^{'}, \ldots, G_{\mathcal{B}}^{'}\}$;
3:     Encode the original graphs and the augmented graphs to get graph representations $h_i$ and $h_i^{'}$;     `// Eq. 2-3`
4:     Calculate the dot product distance between each graph in the batch to get contextual matrix and construct the online global relation graph $G_{\mathcal{B}}$;     `// Eq. 4-5`
5:     Obtain high-order contextual relations from $G_{\mathcal{B}}$ to get contextual feature in affinitive space and retrieve the nearest neighbors $\mathcal{N}^a(i, k)$ for each graph $G_i$;     `// Eq. 6-7`
6:     **for** $G_i, G_i^{'} \in \mathcal{B}$ **do**
7:         Get similarity distribution $Q$ between $G_i$ and $\mathcal{N}^a(i, k)$; get similarity distribution $P$ between $G_i^{'}$ and $\mathcal{N}^a(i, k)$; calculate the KL divergence between $P$ and $Q$ to obtain $\mathcal{L}_{ins}$;     `// Eq. 8-10`
8:         Perform soft clustering for $G_i$ and $\mathcal{N}^a(i, k)$ to obtain $\mathcal{L}_{con}$;     `// Eq. 11`
9:         Calculate the intra-cluster distance $d_i^I$ and nearest-cluster distance $d_i^N$ for $G_i$ to obtain $\mathcal{L}_r$;     `// Eq. 13-15`
10:     **end for**
11:     Update parameters in the model by gradient descent
12: **end while**
13: Obtain $r_i$ for each graph in the test set.
14: **return** $r_i$

---

## B. Proofs

*Proof of Theorem 3.1.* We begin by analyzing $D_{\text{in}}$:

$$
\begin{aligned}
D_{\text{in}} &= \mathbb{E}_{\text{in}} \left( d_i^N(\theta^*) - d_i^I(\theta^*) \right) \\
&= \mathbb{E}_{\text{in}} \left( \|h_i(\theta^*) - \boldsymbol{\mu}_{\phi(h_i)}\|_2 - \|h_i(\theta^*) - \boldsymbol{\mu}_{\phi'(h_i)}\|_2 \right) \\
&\geq \mathbb{E}_{\text{in}} \left( \|h_i(\theta^*) - \boldsymbol{\mu}_{\phi(h_i)}\|_2 - \|h_i(\theta^*) - \boldsymbol{\mu}_{\phi(h_i)}\|_2 \right) \\
&\quad - \mathbb{E}_{\text{in}} \left( \|\boldsymbol{\mu}_{\phi(h_i)} - \boldsymbol{\mu}_{\phi'(h_i)}\|_2 \right).
\end{aligned}
\tag{23}
$$

Similarly, for $D_{\text{out}}$, we have:

$$
\begin{aligned}
D_{\text{out}} &= \mathbb{E}_{\text{out}} \left( d_i^N(\theta^*) - d_i^I(\theta^*) \right) \\
&= \mathbb{E}_{\text{out}} \left( \|h_i(\theta^*) - \boldsymbol{\mu}_{\phi(h_i)}\|_2 - \|h_i(\theta^*) - \boldsymbol{\mu}_{\phi'(h_i)}\|_2 \right) \\
&\leq \mathbb{E}_{\text{out}} \left( \|\boldsymbol{\mu}_{\text{out}} - \boldsymbol{\mu}_{\phi'(h_i)}\|_2 \right) - \mathbb{E}_{\text{out}} \left( \|\boldsymbol{\mu}_{\text{out}} - \boldsymbol{\mu}_{\phi(h_i)}\|_2 \right) \\
&\quad + 2\mathbb{E}_{\text{out}} \left( \|h_i(\theta^*) - \boldsymbol{\mu}_{\text{out}}\|_2 \right).
\end{aligned}
\tag{24}
$$

By the assumption that

$$
\mathbb{E}_{\text{out}} \left( \|\boldsymbol{\mu}_{\text{out}} - \boldsymbol{\mu}_{\phi'(h_i)}\|_2 \right) - \mathbb{E}_{\text{out}} \left( \|\boldsymbol{\mu}_{\text{out}} - \boldsymbol{\mu}_{\phi(h_i)}\|_2 \right)
\tag{25}
$$

is sufficiently smaller than

$$\mathbb{E}_{\text{in}} \left( \| \boldsymbol{\mu}_{\phi(h_i)} - \boldsymbol{\mu}_{\phi'(h_i)} \|_2 \right),\tag{26}$$

such that

$$\mathbb{E}_{\text{out}} \left( \| \boldsymbol{\mu}_{\text{out}} - \boldsymbol{\mu}_{\phi'(h_i)} \|_2 \right) - \mathbb{E}_{\text{out}} \left( \| \boldsymbol{\mu}_{\text{out}} - \boldsymbol{\mu}_{\phi(h_i)} \|_2 \right)$$
$$\leq \mathbb{E}_{\text{in}} \left( \| \boldsymbol{\mu}_{\phi(h_i)} - \boldsymbol{\mu}_{\phi'(h_i)} \|_2 \right) + 2\mathbb{E}_{\text{out}} \left( \| h_i(\theta^*) - \boldsymbol{\mu}_{\text{out}} \|_2 \right),\tag{27}$$

we conclude that $D_{\text{out}} < D_{\text{in}}$. Thus, the theorem is proved. □

To prove Theorem 3.2, we need the following two lemmas.

Define the empirical process $G_n$ as:

$$G_n = \sqrt{n} \frac{1}{n} \sum_{i=1}^{n} \left( r_i(\theta) - \mathbb{E}\left( r_i(\theta) \right) \right),\tag{28}$$

where $r_i(\theta)$ is the compression ratio for the $i$-th graph at parameter $\theta$. Let $\mathcal{F}$ denote a class of measurable functions indexed by the parameters of the neural network with envelope function $F \equiv 2$, since $r_i$ is bounded by 2. We define the $L_2$ norm of $G_n$ with respect to $\mathcal{F}$ as:

$$\|G_n\|_{\mathcal{F}} = \sup_{f \in \mathcal{F}} |\langle G_n, f \rangle|,\tag{29}$$

where $\langle \cdot, \cdot \rangle$ denotes the inner product. Define the entropy integral as:

$$J_{[\,]}(\epsilon, \mathcal{F}, L_2(P)) = \int_0^\epsilon \sqrt{\log N(\epsilon, \mathcal{F}, L_2(P))} \, d\epsilon,\tag{30}$$

in which $N(\epsilon, \mathcal{F}, L_2(P))$ represents the covering number of $\mathcal{F}$ with respect to $L_2(P)$. We then introduce the first lemma.

**Lemma B.1.** *Given any class $\mathcal{F}$ of measurable functions and its envelope function $F$, we have*

$$\mathbb{E}_P^* \|G_n\|_{\mathcal{F}} \lesssim J_{[\,]}\left( \|F\|_{P,2}, \mathcal{F}, L_2(P) \right).\tag{31}$$

To compute the entropy integral, we need to bound the covering number of the class $\mathcal{F}$ with respect to $L_2(P)$. We then introduce the second lemma.

**Lemma B.2.** *A universal constant $K$ exists such that for every VC class $\mathcal{F}$ of functions, every $r \geq 1$ and $0 < \epsilon < 1$,*

$$\sup_Q N\left( \epsilon \|F\|_{Q,r}, \mathcal{F}, L_r(Q) \right) \leq KV(\mathcal{F})(16e)^{V(\mathcal{F})} \left( \frac{1}{\epsilon} \right)^{r(V(\mathcal{F})-1)}.\tag{32}$$

These two lemmas are based on the empirical process theory (Palmes, 2014) and can be found in Van der Vaart (2000).

*Proof of Theorem 3.2.* By Lemmas B.1 and B.2, we have:

$$\frac{\mathbb{E}_P^* \|G_n\|_{\mathcal{F}}}{\sqrt{n}} \lesssim \sqrt{\frac{d_{VC}}{n}}.\tag{33}$$

Given the assumption that $\sum_{G_i} r_i(\hat{\theta}) - \sum_{G_i} r_i(\theta^*) \leq 0$, we have:

$$\begin{aligned}
\mathbb{E}_{\text{in}}\left( r(\hat{\theta}) \right) - \mathbb{E}_{\text{in}}\left( r(\theta^*) \right) &= n^{-1} \sum_{G_i} r_i(\hat{\theta}) - n^{-1} \sum_{G_i} r_i(\theta^*) \\
&\quad + \mathbb{E}_{\text{in}}\left( r(\hat{\theta}) \right) - n^{-1} \sum_{G_i} r_i(\hat{\theta}) \\
&\quad + n^{-1} \sum_{G_i} r_i(\theta^*) - \mathbb{E}_{\text{in}}\left( r(\theta^*) \right) \\
&\leq 2 \frac{\mathbb{E}_P^* \|G_n\|_{\mathcal{F}}}{\sqrt{n}} \lesssim \sqrt{\frac{d_{VC}}{n}}.
\end{aligned}\tag{34}$$

Thus, we have the desired result. □

## C. Implement Details

All experiments are conducted on an NVIDIA RTX4090 GPU with 24GB memory. We implement our model based on PyTorch and PyTorch Geometric. We utilize a random splitting strategy to ensure diversity. Specifically, following previous research (Liu et al., 2023; He et al., 2024), 90% of ID examples are used for training, and 10% of ID examples and the same numbers of OOD examples are integrated for testing. For training, we utilized the Adam optimizer with an initial learning rate of $1e^{-3}$. The batch size is set to 128, and the maximum epoch is set to 400. More concretely, for the method-specific hyperparameters, $\beta$ and $\gamma$ are fixed to 0.3 and 1.0, respectively, across all dataset pairs. The remaining hyperparameters are selected from predefined candidate sets, where $k_1 \in \{5, 10, 15\}$, $k_2 \in \{3, 5\}$, $k \in \{3, 10, 15\}$, and $C \in \{3, 5\}$.

## D. Performance Comparison on Anomaly Detection

*Table 3.* The compared performance in terms of AUC (in percent).

| Method | PK-OCSVM | PK-iF | WL-OCSVM | WL-iF | InfoGraph-iF | GraphCL-iF | OCGIN | GLocalKD | GOOD-D | HGOE | CLINIC |
|---|---|---|---|---|---|---|---|---|---|---|---|
| PROTEINS-full | $50.49_{\pm4.92}$ | $60.70_{\pm2.55}$ | $51.35_{\pm4.35}$ | $61.36_{\pm2.54}$ | $57.47_{\pm3.03}$ | $60.18_{\pm2.53}$ | $70.89_{\pm2.44}$ | $\underline{77.30_{\pm5.15}}$ | $71.97_{\pm3.86}$ | $73.13_{\pm0.46}$ | $\mathbf{77.82_{\pm3.50}}$ |
| ENZYMES | $53.67_{\pm2.66}$ | $51.30_{\pm2.01}$ | $55.24_{\pm2.66}$ | $51.60_{\pm3.81}$ | $53.80_{\pm4.50}$ | $53.60_{\pm4.88}$ | $58.75_{\pm5.98}$ | $61.39_{\pm8.81}$ | $63.90_{\pm3.69}$ | $\underline{67.28_{\pm0.99}}$ | $\mathbf{68.08_{\pm5.78}}$ |
| AIDS | $50.79_{\pm4.30}$ | $51.84_{\pm2.87}$ | $50.12_{\pm3.43}$ | $61.13_{\pm0.71}$ | $70.19_{\pm5.03}$ | $79.72_{\pm3.98}$ | $78.16_{\pm3.05}$ | $93.27_{\pm4.19}$ | $97.28_{\pm0.69}$ | $\underline{97.84_{\pm0.55}}$ | $\mathbf{98.56_{\pm0.49}}$ |
| DHFR | $47.91_{\pm3.76}$ | $52.11_{\pm3.96}$ | $50.24_{\pm3.13}$ | $50.29_{\pm2.77}$ | $52.68_{\pm3.21}$ | $51.10_{\pm2.35}$ | $49.23_{\pm3.05}$ | $56.71_{\pm3.57}$ | $62.67_{\pm3.11}$ | $64.39_{\pm0.68}$ | $\mathbf{65.95_{\pm4.20}}$ |
| BZR | $46.85_{\pm5.31}$ | $55.32_{\pm6.18}$ | $50.56_{\pm5.87}$ | $52.46_{\pm3.30}$ | $63.31_{\pm8.52}$ | $60.24_{\pm5.37}$ | $65.91_{\pm1.47}$ | $69.42_{\pm7.78}$ | $75.16_{\pm5.15}$ | $\mathbf{80.54_{\pm1.35}}$ | $\underline{77.02_{\pm6.22}}$ |
| COX2 | $50.27_{\pm7.91}$ | $50.05_{\pm2.06}$ | $49.86_{\pm7.43}$ | $50.27_{\pm0.34}$ | $53.36_{\pm8.86}$ | $52.01_{\pm3.17}$ | $53.58_{\pm5.05}$ | $59.37_{\pm12.67}$ | $62.65_{\pm8.14}$ | $\mathbf{69.52_{\pm2.68}}$ | $\underline{65.70_{\pm5.05}}$ |
| DD | $48.30_{\pm3.98}$ | $71.32_{\pm2.41}$ | $47.99_{\pm4.09}$ | $70.31_{\pm1.09}$ | $55.80_{\pm1.77}$ | $59.32_{\pm3.92}$ | $72.27_{\pm1.83}$ | $\underline{80.12_{\pm5.24}}$ | $73.25_{\pm3.19}$ | $76.95_{\pm2.24}$ | $\mathbf{81.91_{\pm2.81}}$ |
| NCI1 | $49.90_{\pm1.18}$ | $50.58_{\pm1.38}$ | $50.63_{\pm1.22}$ | $50.74_{\pm1.70}$ | $50.10_{\pm0.87}$ | $49.88_{\pm0.53}$ | $\mathbf{71.98_{\pm1.21}}$ | $\underline{68.48_{\pm2.39}}$ | $61.12_{\pm2.21}$ | $65.82_{\pm1.43}$ | $67.06_{\pm1.94}$ |
| IMDB-B | $50.75_{\pm3.10}$ | $50.80_{\pm3.17}$ | $54.08_{\pm5.19}$ | $50.20_{\pm0.40}$ | $56.50_{\pm3.58}$ | $56.50_{\pm4.90}$ | $60.19_{\pm8.90}$ | $52.09_{\pm3.41}$ | $65.88_{\pm0.75}$ | $\underline{69.82_{\pm1.37}}$ | $\mathbf{73.14_{\pm4.07}}$ |
| REDDIT-B | $45.68_{\pm2.24}$ | $46.72_{\pm3.42}$ | $49.31_{\pm2.33}$ | $48.26_{\pm0.32}$ | $68.50_{\pm5.56}$ | $71.80_{\pm4.38}$ | $75.93_{\pm8.65}$ | $77.85_{\pm2.62}$ | $88.67_{\pm1.24}$ | $\underline{89.41_{\pm1.21}}$ | $\mathbf{89.92_{\pm1.25}}$ |
| HSE | $57.02_{\pm8.42}$ | $56.87_{\pm10.51}$ | $62.72_{\pm10.13}$ | $53.02_{\pm5.12}$ | $53.56_{\pm3.98}$ | $51.18_{\pm2.71}$ | $64.84_{\pm4.70}$ | $59.48_{\pm1.44}$ | $\underline{69.65_{\pm2.14}}$ | $\mathbf{74.50_{\pm3.73}}$ | $66.50_{\pm1.21}$ |
| MMP | $46.65_{\pm6.31}$ | $50.06_{\pm3.73}$ | $55.24_{\pm3.26}$ | $52.68_{\pm3.34}$ | $54.59_{\pm2.01}$ | $54.54_{\pm1.86}$ | $\underline{71.23_{\pm0.16}}$ | $67.84_{\pm0.59}$ | $70.51_{\pm1.56}$ | $\mathbf{71.94_{\pm0.54}}$ | $70.27_{\pm1.30}$ |
| p53 | $46.74_{\pm4.88}$ | $50.69_{\pm2.02}$ | $54.59_{\pm4.46}$ | $50.85_{\pm2.16}$ | $52.66_{\pm1.95}$ | $53.29_{\pm2.32}$ | $58.50_{\pm0.37}$ | $64.20_{\pm0.81}$ | $62.99_{\pm1.55}$ | $64.70_{\pm1.16}$ | $\mathbf{66.41_{\pm0.35}}$ |
| PPAR-gamma | $53.94_{\pm6.94}$ | $45.51_{\pm2.58}$ | $57.91_{\pm6.13}$ | $49.60_{\pm0.22}$ | $51.40_{\pm2.53}$ | $50.30_{\pm1.56}$ | $71.19_{\pm4.28}$ | $64.59_{\pm0.67}$ | $67.34_{\pm1.71}$ | $\underline{71.92_{\pm4.17}}$ | $\mathbf{72.10_{\pm0.91}}$ |
| Avg. Rank | 9.7 | 8.8 | 8.1 | 8.6 | 7.4 | 7.6 | 4.6 | 4.1 | 3.4 | $\underline{2.0}$ | **1.6** |

Following the benchmark established in (Ma et al., 2022), we conduct anomaly detection experiments to investigate whether CLINIC can generalize to this task, with results presented in Table 3. Our findings indicate that: 1) Consistent with the results in Table 1, end-to-end approaches outperform two-step approaches, while non-deep two-step methods demonstrate the weakest performance. 2) CLINIC also shows strong competitiveness in anomaly detection, highlighting its promising capability to capture the contextual patterns of normal data through the leveraging of semantic neighborhoods.

## E. Ablation Study on Consistency Loss

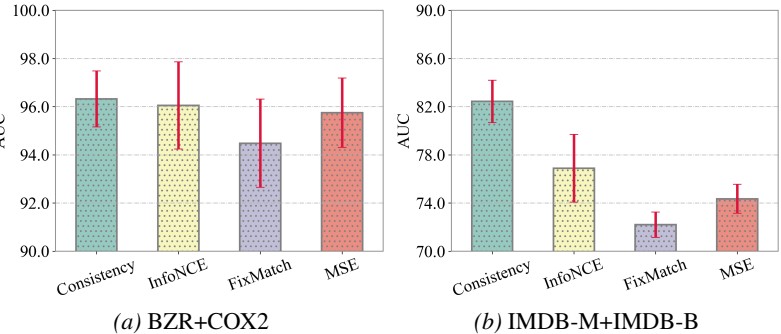

*(a)* BZR+COX2                    *(b)* IMDB-M+IMDB-B

*Figure 7.* Ablation study on different consistency loss.

To examine the role of the consistency loss, we conduct comparative experiments between $\mathcal{L}_{ins}$ and several widely adopted alternatives, including contrastive losses such as InfoNCE and mean squared error (MSE), as well as the consistency-based FixMatch loss. The results in Figure 7 indicate that the proposed consistency loss further surpasses InfoNCE and MSE, suggesting that matching the similarity distributions between the original graph and its augmented counterpart can produce more transferable and robust representations than directly imposing instance-level cross-view similarity constraints.

## F. Complexity Analysis

The computational consumption of our model is mainly composed of three parts: (i) contextually affinitive neighborhood retrieval; (ii) consistency regularization; (iii) decision boundary compression. Given the graph with average number of nodes and edges $|\mathcal{V}|$ and $|\mathcal{E}|$, the number of encoder layer is $L$, the representation dimension is $d'$, and the batch size is $|\mathcal{B}|$. For (i), the time complexity of the GNN-based encoder is $O(|\mathcal{E}|Ld')$, the time complexity of the construction of global contextual $k$-NN graph is $O(|\mathcal{B}|\log k_1 + k_1 \log k_2)$, the time complexity of the $k$ nearest affinitive neighborhood retrieval is $O(|\mathcal{B}|(k_2 L + \log k))$. For (ii), we perform instance-aware consistency regularization from two views, which takes $O(2kd')$ for each graph. Meanwhile, given $C$ clusters in the task, the time complexity of contextual-aware consistency regularization for each graph is $O(Cd'(k+1))$. For (iii), the time complexity is $O(2d')$ for each graph. In total, the overall time complexity of CLINIC is $O(|\mathcal{E}|Ld' + |\mathcal{B}|\log k_1 + k_1 \log k_2 + |\mathcal{B}|(k_2 L + \log k) + Cd'(k+1) + 2d')$, which is linearly related to the graph size in each graph instance. This is computationally comparable to other graph OOD methods.

