# OpenReview forum: "CLINIC: Towards High-quality Graph Out-Of-Distribution Detection"
_ICML.cc/2026/Conference — ICML 2026 regular_

### Official Review · Reviewer_V9mv · 2026-03-07

**Soundness:** 3
**Presentation:** 3
**Significance:** 3
**Originality:** 3
**Overall Recommendation:** 4
**Confidence:** 4

**Summary:**

This paper proposes a graph OOD detection framework that identifies anomalies by exploring semantic structures within contextual affinity neighborhoods rather than relying solely on topological patterns. It employs twin concordance learning to ensure representation consistency and a decision boundary compression strategy to improve the separation between ID and OOD samples.

**Compliance With Llm Reviewing Policy:**

Affirmed.

**Final Justification:**

Thanks for the response. My concerns are mostly solved, and I would raise my score to 4.

But I still want to see the details of the joint online learning process (e.g., formula, theoretical proof and so on). This work shares some similar ideas with JEPA's from LeCun, which both emphasize the importance of the alignment within latent space. To me, how to create a meaningful shared latent space first is an interesting question. The score would be further increased If the authors can provide further details.

**Key Questions For Authors:**

See weaknesses above.

**Limitations:**

yes

**Strengths And Weaknesses:**

Stengths:

1. The research problem of graph OOD detection is an interesting and important question.

2. The idea of contextual affinity is quite interesting to me.

3. The experiments are quite extensive.

Weaknesses:

1. It's ambiguous how the GNN encoder (illustrated from line 152 to line 154 in Section 3.2) guarantee the assumption that the more high-order contextual information two graphs share, the more similar the corresponding graph data points are in the transformed space. It's the foundation of the proposed contextual affinity space, but I don't see the training detail, theoretical proof or ablation experiment to prove this assumption.

2. It's also unclear how the graph data augmentation strategies, illustrated from line 191 to line 192 in Section 3.3, guarantee that the generated graph view $G_i^{'}$ preserves the intrinsic structural and attribute information of $G_i$. If there are too many distribution shifts between $G_i$ and $G_i^{'}$, it doesn't make sense to enhance the consistency of these two graphs through twin concordance learning.

3. Figure 2 needs to be improved, as it lacks narrative quality. In fact, I don't think it's aligned with the text below, and I have to read the text below so that I can understand what CLINIC is about.

Note:

I might give you a better grade if you can address my concerns carefully.

---

> ### Author Rebuttal · Authors · 2026-03-30
>
> We are truly thankful for your insightful and constructive review. Our detailed responses are presented below.
>
> > Q1. The foundation of the contextual affinity space.
>
> **R1：** Thanks for your valuable comment! To illustrate the intuition of the assumption, we consider a toy example consisting of $2n$ points in a $p$-dimensional space. These points are divided into two groups: $\{\mathbf{x}_1, \dots, \mathbf{x}_n\}$ and $\{\mathbf{y}_1, \dots, \mathbf{y}_n\}$, with mean vectors denoted by $\mathbf{\mu}_x$ and $\mathbf{\mu}_y$, respectively. Each observed point is generated as $\mathbf{x}_i = \mathbf{\mu}_x + \mathbf{\epsilon}_i, \quad \mathbf{y}_j = \mathbf{\mu}_y + \mathbf{\epsilon}_j$, where the error terms $\mathbf{\epsilon}_i$ and $\mathbf{\epsilon}_j$ are independent standard normal random vectors. The Euclidean distance in the original space is given by $d(\mathbf{x}_i, \mathbf{y}_j) = \|\mathbf{x}_i - \mathbf{y}_j\|_2$. By the concentration of measure phenomenon [1], we have approximately:
> $$\mathbb{E}[d(\mathbf{x}_i, \mathbf{y}_j)] \approx \|\mathbf{\mu}_x - \mathbf{\mu}_y\|_2 + C\sqrt{p}$$
> where $C$ is a positive constant.
>
> **In contrast, we consider computing distances in a contextual affinity space, where each point is smoothed by averaging with its nearest neighbors.** Specifically, let the $K$ nearest neighbors of $\mathbf{x}_ i$ be $\mathbf{x}_ {i_1}, \dots, \mathbf{x}_ {i_ K}$, and the $K$ nearest neighbors of $\mathbf{y}_ j$ be $\mathbf{y}_ {j_ 1}, \dots, \mathbf{y}_ {j_ K}$. The affinity features are then defined as the neighborhood averages:
> $$\mathbf{\mu}_ {\mathbf{x}_ i} = \frac{1}{K}\sum_ {k=1}^ K \mathbf{x}_ {i_ k}, \quad \mathbf{\mu}_ {\mathbf{y}_ j} = \frac{1}{K}\sum_ {k=1}^ K \mathbf{y}_ {j_k}$$
>
> The distance in this affinity space is $d(\mathbf{\mu}_ {\mathbf{x}_ i}, \mathbf{\mu}_ {\mathbf{y}_ j}) = \|\mathbf{\mu}_ {\mathbf{x}_ i} - \mathbf{\mu}_ {\mathbf{y}_ j}\|_ 2$. Again, applying the concentration of measure phenomenon, we obtain:$$\mathbb{E}[d(\mathbf{\mu}_ {\mathbf{x}_ i}, \mathbf{\mu}_ {\mathbf{y}_ j})] \approx \|\mathbf{\mu}_ x - \mathbf{\mu}_ y\|_2 + C\sqrt{\frac{p}{K}}$$
>
> This rigorously proves that as the $K$ increases, the noise term $C\sqrt{p/K}$ diminishes significantly. **Consequently, the graphs mathematically converge towards their true semantic distance, becoming highly similar in the transformed space.**
>
> > Q2. The potential risk of distribution shift.
>
> **R2:** Thanks for your insightful comment! The stochastic augmentations are controlled by predefined probability ratios of 0.5, ensuring localized perturbations rather than severe distribution shifts. More importantly, while traditional contrastive methods rely solely on the exact alignment of $G_i$ and $G_i^{'}$, CLINIC utilizes contextual affinity. **Even if $G_i^{'}$ undergoes a slight structural deformation, its higher-order relationships with the stable contextual neighborhood remain intact.** This can also be proved by R1 above. In traditional contrastive learning, the expected distance is dominated by the noise term:
>
> $$ \mathbb{E}[d(\mathbf{x}_i, \mathbf{y}_j)] \approx \|\mathbf{\mu}_x - \mathbf{\mu}_y\|_2 + C\sqrt{p},$$
>
> This aligns with your concern regarding distribution shifts. **However, CLINIC aligns graphs using contextual affinity. By smoothing the representations over $K$ nearest contextual neighbors, the affinity feature becomes a neighborhood average.** Theoretically, the expected distance in this affinity space becomes:
>
> $$ \mathbb{E}[d(\mathbf{\mu}_ {\mathbf{x}_ i}, \mathbf{\mu}_ {\mathbf{y}_ j})] \approx \|\mathbf{\mu}_ x - \mathbf{\mu}_ y\|_ 2 +C\sqrt{\frac{p}{K}}.$$
>
> This demonstrates that the structural noise introduced by augmentations is suppressed by a factor of $1/\sqrt{K}$. Even if $G_i^{'}$ undergoes structural deformation, its aggregated higher-order relationships heavily filter out the augmentation noise.
>
> We further perform experiments to examine the impact of augmentation intensity controlled by varying perturbation ratios. Results listed in the following table indicate that CLINIC is not sensitive to the augmentation intensity.
> | Augmentation Ratio | Tox21+SIDER | BBBP+BACE|
> | :-: | :-: | :-: |
> | 0.1 | 70.43 | 83.27 |
> | 0.3 | 70.57 | 83.72 |
> | 0.5 | 70.69 | 84.21 |
> | 0.7 | 70.04 | 83.67 |
> | 0.9 | 68.85 | 82.45 |
>
> > Q3. Figure 2 needs to be improved.
>
> **R3:** Thanks for your valuable comment! We modified Figure 2 in https://anonymous.4open.science/r/CLINIC-for-OOD-detection-F8C1/framework.jpg. We will provide a more clear illustration in the revised paper.
>
> **Reference:**
>
> [1] High-dimensional probability: An introduction with applications in data science. 2018.
>
> In light of these responses, we hope we have addressed your concerns, and hope you will consider raising your score. We will properly include all the rebuttal contents in the revised version, following your valuable suggestions.

---

> > ### Author Rebuttal · Reviewer_V9mv · 2026-04-01
> >
> > Thanks for your detailed response. My questions are partially resolved. My follow-up questions for the authors are as follows:
> >
> > Regarding **W1**, it seems that you misunderstood my question. What I truly want to ask is how you guarantee that the graph representation $\mathcal{H}$ contains the clean information, that can support your assumption. Do you use a pre-trained GNN, or use a jointly-trained ones along with your further training process?
> >
> > Regarding **W2 & W3**, the authors have resolved my major concerns.
> >
> > Due to the remaining question, I would like to keep my score and look forward to the following response from the authors.

---

> > > ### Author Response · Authors · 2026-04-02
> > >
> > > Thanks for your feedback! We are pleased to address your further questions as follows:
> > > > Q: How to guarantee that the graph representation
> > > $\mathcal{H}$ contains the clean information, that can support the assumption?
> > >
> > > Thanks for your question. To validate the effectiveness of our contextual affinity space, we introduce a model variant which replaces this with the space from the graph edit distance. The compared results are shown as below.
> > >
> > > | Method | BZR+COX2 | PTC-MR+MUTAG |
> > > | :--- | :---: | :---: |
> > > | Ours w/ graph edit distance | 90.18 | 78.46 |
> > > | Ours | 96.32 | 82.60 |
> > >
> > > From the results, we can observe the effectiveness of the contextual affinity space, which empircally validates our assumption. As for details, the GNN encoder is jointly learned online in an unsupervised manner with Twin Concordance Learning. We will include all the above results in the revised version.
> > >
> > > Thank you again for your feedback! We will add the rebuttal contents to the main paper in the final version following your valuable suggestions. Please let us know if you have further questions.

---

### Official Review · Reviewer_KA4h · 2026-03-07

**Soundness:** 3
**Presentation:** 3
**Significance:** 3
**Originality:** 3
**Overall Recommendation:** 4
**Confidence:** 4

**Summary:**

This paper proposes CLINIC, a graph-level out-of-distribution (OOD) detection method that emphasizes semantic structure rather than only topological similarity. The key idea is to build a contextual affinity graph within each minibatch using reciprocal nearest-neighbour relations, then apply a meta-GNN over that affinity graph to obtain high-order contextual relations for neighbour retrieval. On top of this, CLINIC introduces twin concordance learning: a perturbation-aware term that aligns similarity distributions between an original graph and its augmentation using symmetric KL, and a contextual-aware clustering consistency term that encourages neighbours to share cluster assignments. It applies a decision boundary compression objective to pull ID representations toward their cluster centers and uses the resulting compression ratio as the OOD score at test time. Experiments on multiple datasets indicates improvements over several OOD baselines and show ablations indicating contributions from each loss component.

**Compliance With Llm Reviewing Policy:**

Affirmed.

**Final Justification:**

The authors' response has adequately addressed all my questions. I would like to maintain my positive score.

**Key Questions For Authors:**

1. While the computational cost is analysed in the Appendix F, detailed statistics in terms of memory usage, training/inference time comparison is suggested to be included.
2. Have the authors considered other detection metrics like AUPR, FPR95 to better illustrate the methods efficacy under constrained situations?
3. Have the authors considered evaluations on near-ID scenarios?
4. How does CLINIC behave if the contextual affinity is computed from a fixed reference memory bank rather than the current minibatch?
5. What is the sensitivity to batch size and batch composition (class imbalance, dataset size)?

**Limitations:**

yes

**Strengths And Weaknesses:**

Strengths:
1. The authors provide a meaningful motivation on why Euclidean neighbourhood similarity can be misleading across datasets and motivates contextual affinity as capturing the same distribution semantics beyond raw topology.
2. The proposed method and pipeline are clear and coherent.
3. The experiments are solid and a comprehensive analysis of different hyperparameters, ablation, distribution visualisation was provided to justify the effectiveness of CLINIC.

Weaknesses:
1. Even if edges are sparse, constructing similarity over a batch and kNN retrieval can result in O(|B|^2 d) similarity computation unless approximated? The paper's complexity discussion does not clearly account for the potential of dense similarity matrix computation cost. Without wall-clock time and GPU memory use comparisons, it is hard to validate the compute claim.
2. The analysis in line 362 notes that too many neighbours can introduce noise and degrade performance (false positives). But the method’s losses directly encourage neighbours to agree in cluster assignment. If the retrieval is wrong early in training, concordance can amplify errors, collapsing nearby but actually distinct modes into one cluster and harming boundary quality.
3. Since the OOD score is the compression ratio r derived from distances to the assigned cluster center and the nearest alternative center, decision boundary compression can mis-score OOD that lies near an ID cluster. For such cases, d_I  can be small and r_i can also be small, causing false negatives.

---

> ### Author Rebuttal · Authors · 2026-03-30
>
> We are truly grateful for the time you have taken to review our paper, your insightful comments, and support.
>
> > Q1. Comparisons on wall-clock time and GPU memory.
>
> **R1:** Thanks for your valuable comment! Please refer to R2 in Reviewer Ldrb.
>
> > Q2. Concern about FP caused by error accumulation in early training.
>
> **R2:** Thanks for your valuable comment! Our CLINIC has incorporated two mechanisms to prevent possible degradation. First, we introduce perturbation-aware loss that operates at the instance level. In the early training stages, when neighbor retrieval is inevitably noisy, this instance-level consistency acts as a robust regularization anchor. It stabilizes the latent space and resists arbitrary mode collapse before meaningful contextual semantics emerge. Second, our contextual concordance utilizes soft clustering assignments rather than hard pseudo-labels, allowing the network to self-correct the affinity graph as representation quality improves. We conducted experiments and reported the total number of FP under our setting and variant (removal of concordance learning).
>
> | Method | BZR+COX2 | AIDS+DHFR | IMDB-M+IMDB-B
> | :- | :-: | :-: | :-: |
> | variant | 12 | 6 | 106 |
> | Ours | 3 | 0 | 81 |
>
> > Q3. Concern about FP caused by OOD score.
>
> **R3:** Thanks for your valuable comment! Unlike standard Euclidean spaces, our latent space is shaped by contextual affinity and twin concordance learning. The ID cluster centers represent highly dense regions of specific high-order semantic relationships inherent to the ID distribution. Because OOD graphs originate from different distributions, they lack these specific structural and contextual affinities. Consequently, the network naturally projects them away from the ID cluster centers. We also calculated the counts of FP in the following table.
> | Method | BZR+COX2 | IMDB-M+IMDB-B
> | :- | :-: | :-: |
> | GOOD | 8 | 87 |
> | HGOE | 6 | 92 |
> | Ours | 3 | 81 |
>
> > Q4. Using metrics like AUPR, FPR95.
>
> **R4:** Thanks for your valuable comment! We reported the results on AUPR and FPR95 in the following table.
> | Metrics | BZR+COX2 | PTC-MR+MUTAG | AIDS+DHFR |
> | :- | :-: | :-: | :-: |
> | Ours (AUPR) | 0.92 | 0.76 | 0.99 |
> | Ours (FPR95) | 0.15 | 0.31 | 0.09 |
> | GOOD (AUPR) | 0.92 | 0.74 | 0.99 |
> | GOOD (FPR95) | 0.14 | 0.28 | 0.10 |
> | HGOE (AUPR) | 0.91 | 0.75 | 0.99 |
> | HGOE (FPR95) | 0.17 | 0.30 | 0.09 |
>
> > Q5. Have the authors considered evaluations on near-ID scenarios?
>
> **R5:** Thanks for your valuable comment! As detailed in Section 4.1, we construct evaluation pairs from identical domains (e.g., BBBP and BACE for molecular graphs, IMDB-M and IMDB-B for social networks) that share similar feature spaces but exhibit subtle distribution shifts. These pairs are widely recognized as highly challenging near-ID benchmarks. Our superior performance in Table 1 demonstrates CLINIC's strong capability in near-ID settings.
>
> > Q6. Replace minibatch with memory bank.
>
> **R6:** Thanks for your valuable comment! The introduction of a fixed reference memory bank may face the challenge of feature staleness. The representation stored in the memory bank would inevitably lag behind the current encoder's latest state, leading to the retrieval of incorrect neighbors, thereby injecting destructive noise into our twin concordance learning. In contrast, while our current minibatch-based strategy operates with a smaller candidate pool, it strictly guarantees real-time synchronization and absolute matching of all features, avoiding staleness errors with lower memory overhead. The results of the ablation study listed in the following table demonstrate our statement.
> | Metrics | BZR+COX2 | PTC-MR+MUTAG | AIDS+DHFR |
> | :- | :-: | :-: | :-: |
> | Memory Bank | 94.85 | 78.24 | 99.10 |
> | Ours | 96.32 | 82.60 | 99.42 |
>
> > Q7. What is the sensitivity to batch size and batch composition?
>
> **R7:** Thanks for your valuable comment! We conducted experiments by varying batch size (B) and class imbalance ratio (R, where we define $R=N_1/N_C$, where $N_1\geq{N_2}\geq{...}\geq{N_C}$). The results are listed in the following table. Regarding batch size, CLINIC remains robust across moderate sizes. In our experimental setting, we set B=128. Regarding batch composition and class imbalance, the soft cluster assignment mitigates the impact of class imbalance. This probabilistic approach provides gradient elasticity, ensuring that minority-class graphs preserve their distinct semantic boundaries and are not overwhelmed by the majority class within an imbalanced batch.
>
> | Method | B=32 | B=64 | B=128 | B=256 | R=0.5 | R=0.6 | R=0.7|
> | :- | :-: | :-: | :-: | :-: | :-: | :-: | :-: |
> | BZR+COX2 | 96.20 | 96.28 | 96.32 | 96.30 | 96.31 | 96.05 | 95.61 |
> | AIDS+DHFR | 99.48 | 99.40 | 99.42 | 99.31 | 99.42 | 98.81 | 98.22 |
> | Tox21+SIDER | 70.61  | 70.68 | 70.69 | 70.63 | 70.52 | 70.30 | 69.84 |
>
> We will properly include all the rebuttal contents in the revised version, following your valuable suggestions.

---

> > ### Author Rebuttal · Reviewer_KA4h · 2026-04-01
> >
> > Thanks to the author for the reply. I have no more questions and will keep my score. Good Luck.

---

> > > ### Author Response · Authors · 2026-04-02
> > >
> > > Thanks for your positive feedback and for confirming that our rebuttal addressed your concerns! We will properly include all the rebuttal concerns in the revised version, following your valuable suggestions.

---

### Official Review · Reviewer_7Zrh · 2026-03-10

**Soundness:** 3
**Presentation:** 3
**Significance:** 3
**Originality:** 3
**Overall Recommendation:** 4
**Confidence:** 3

**Summary:**

This paper studies graph out-of-distribution (OOD) detection, which aims to identify graphs that do not belong to the training distribution. The authors argue that existing graph OOD detection methods mainly rely on topological structure similarities while overlooking the semantic structure preserved in contextual affinity neighborhoods.
To address this issue, the paper proposes CLINIC (Contextual Affinity Exploration with Twin Concordance). The method first constructs a contextual affinity graph to capture semantic relationships among graph samples. A meta-graph neural network is then used to model higher-order affinities. To improve representation robustness, the method introduces twin concordance learning, which enforces consistency between graph views and their contextual neighbors. Finally, a decision boundary compression mechanism is used to enhance separation between in-distribution (ID) and out-of-distribution (OOD) graphs. Experiments on multiple benchmark datasets show that CLINIC achieves improved OOD detection performance compared to existing baselines.

**Compliance With Llm Reviewing Policy:**

Affirmed.

**Final Justification:**

Thank the authors for their responses, which resolved most of my confusion, so I raised my score (3 -> 4).

**Key Questions For Authors:**

1. Among contextual affinity modeling, twin concordance learning, and boundary compression, which component contributes most to the performance gains?

2. What is the computational complexity of constructing the contextual affinity graph and retrieving neighbors when scaling to larger graph datasets?

3. The method relies on several hyperparameters (e.g., number of clusters and neighbors). How sensitive is the model to these choices?

**Limitations:**

The paper could further discuss limitations such as computational overhead of affinity graph construction, sensitivity to hyperparameters, and scalability to large graph datasets.

**Strengths And Weaknesses:**

### Strengths

1. The proposed method is technically reasonable and supported by empirical experiments on multiple datasets. The framework includes ablation studies and parameter analysis that help demonstrate the contributions of different components.

2. The paper clearly motivates the importance of leveraging semantic relationships in graph OOD detection. The method pipeline is reasonably structured and the experimental setup is described in sufficient detail.

3. The paper introduces the idea of contextual affinity exploration combined with concordance learning for graph OOD detection, and integrates these ideas with a decision boundary compression strategy.

### Weaknesses

1. The method involves several components (affinity graph construction, concordance learning, and boundary compression), and it is not entirely clear which component contributes most to the final performance improvements.

2. Some technical details of contextual affinity construction and high-order affinity modeling could be explained more clearly to improve readability.

3. Several components of the framework build upon existing ideas such as contrastive learning, consistency regularization, and clustering-based separation, and the novelty mainly lies in their combination.

---

> ### Author Rebuttal · Authors · 2026-03-30
>
> We are truly grateful for the time you have taken to review our paper and your insightful review. Here we address your comments in the following.
>
> > Q1. The method involves several components (affinity graph construction, concordance learning, and boundary compression), and it is not entirely clear which component contributes most to the final performance improvements.
>
> **R1:** Thanks for your valuable comment! Based on our ablation study, the decision boundary compression strategy contributes most significantly to the performance gains. By dynamically minimizing the ratio of the intra-cluster distance to the nearest inter-cluster distance, it drastically compresses the feature space of ID samples. This establishes a highly compact and strict boundary, effectively filtering out OOD samples.
>
> > Q2. Some technical details of contextual affinity construction and high-order affinity modeling could be explained more clearly to improve readability.
>
> **R2:** Thanks for your valuable comment! Contextual affinity refers to the deep semantic similarity of graph samples in the latent space, derived from high-order neighborhood relationships. Our core motivation is to overcome the over-reliance of existing methods on topologies by leveraging high-order contexts to capture more robust ID semantic patterns. To enhance readability, we will refine Section 3.2 by adding a more detailed technical demonstration in the revised manuscript.
>
> > Q3. Concern about novelty.
>
> **R3:** Thanks for your valuable comment! The novelty of our proposed method lies in three points:
> - **Novel concept:** We propose a paradigm shift from traditional topology-based OOD detection to a contextual affinity perspective.
> - **Novel implementation:** Unlike standard instance-level contrastive learning, we design twin concordance learning to enforce consistency across soft semantic clusters. Additionally, our decision boundary compression provides a mathematically grounded scoring mechanism exclusively tailored for this affinity space.
> - **A unified framework:** We organically integrate these non-trivial components into a unified framework for graph OOD detection， which can achieve superior performance in OOD and anomaly detection tasks.
>
> > Q4. What is the computational complexity of constructing the contextual affinity graph and retrieving neighbors when scaling to larger graph datasets?
>
> **R4:** Thanks for your valuable question. Given the batch size $|\mathcal{B}|$ and a samll number of retrieved neighbors $k_1$, $k_2$ and $k$ ($k_1, k_2, k \ll B$), the overall complexity is linearly related to the product of average graph size and $|\mathcal{B}|^2$, maintaining the same level of computational complexity as GOOD [1]. Specifically, the total complexity of contextual affinity graph construction and meta-GNN is linearly related to the product of average graph size and $|\mathcal{B}|^2$. The complexity of twin concordance learning is linearly related to $k$, while the complexity of decision boundary compression is linearly related to $|\mathcal{B}|$. Both time and space complexity are maintained at the same level of computational complexity as GOOD and HGOE. Furthermore, we have provided the training time of our CLINIC to further demonstrate its computational efficiency in comparison to other baselines. Specifically, we introduce two new dataset pairs: ogbg-molhiv+ogbg-molpcba (with 41,127 and 437,929 samples respectively) and REDDIT-MULTI-5K+REDDIT-BINARY (with avg. node/edge 508.5/594.9 and 429.6/497.8 respectively). The results indicate that the framework maintains an affordable level of complexity in our model.
> | Method | BZR+COX2 (s) | BBBP+BACE (s) | AIDS+DHFR (s) | HIV+PCBA (s) | REDDIT-M+REDDIT-B (s) |
> | :--- | :---: | :---: | :---: | :---: | :---: |
> | GOOD | 53 | 407 | 424 | 3146 | 412 |
> | HGOE | 57 | 429 | 445 | 3850 | 433 |
> | Ours | 54 | 414 | 430 | 3427 | 416 |
>
> > Q5. How sensitive is the model to hyperparameter choices?
>
> **R5:** Thanks for your valuable comment! As detailed in Section 4.4, CLINIC is robust and not highly sensitive to these hyperparameters within reasonable ranges. Through a grid search {2, 3, 5, 10, 15, 20}, we found the optimal number of neighbors generally falls around {3, 5}. The optimal cluster count scales with the dataset size. Furthermore, the loss balancing coefficients are set to 1. Detailed hyperparameter settings for all datasets are provided in Appendix Table 3, demonstrating our framework's stability without requiring exhaustive, dataset-specific tuning.
>
> In light of these responses, we hope we have addressed your concerns, and hope you will consider raising your score. If there are any additional notable points of concern that we have not yet addressed, please do not hesitate to share them, and we will promptly attend to those points.

---

> > ### Author Rebuttal · Reviewer_7Zrh · 2026-04-03
> >
> > Thank the authors for their responses, which resolved most of my confusion, so I raised my score (3 -> 4).

---

> > > ### Author Response · Authors · 2026-04-03
> > >
> > > Thanks for your feedback and for increasing the rating! We will properly include all the rebuttal contents in the revised version, following your valuable suggestions.

---

### Official Review · Reviewer_Ldrb · 2026-03-12

**Soundness:** 3
**Presentation:** 3
**Significance:** 3
**Originality:** 3
**Overall Recommendation:** 5
**Confidence:** 4

**Summary:**

This paper proposes a novel framework for graph OOD detection, termed CLINIC, which leverages the contextual affinity of the samples for discriminative graph representations. It builds a contextual affinity graph based on sample’s similarity and performs twin concordance learning to foster a discriminative decision boundary. Finally, a novel proposed compression strategy is applied to refine the OOD detection. Experiments over OOD detection and anomaly detection against most recent baselines show clear improvements.

**Compliance With Llm Reviewing Policy:**

Affirmed.

**Final Justification:**

Given the rebuttal, I would like to keep my score.

**Key Questions For Authors:**

1. How would the model perform if method replaced the contextually affinitive neighbors used in the loss calculation with neighbors retrieved from the original Euclidean space?
2. What is the overall computational cost of the proposed method? A thorough analysis of its training and inference costs is necessary to evaluate its practicality and to understand the trade-offs between its sophisticated design and computational efficiency.
3. The proposed framework integrates three distinct modules. Among these, which one contributes most significantly to the observed performance gains?
4. The second best result is marked with an underline in Table 1, but there is a duplication of the underline.

**Limitations:**

yes

**Strengths And Weaknesses:**

### Strengths
1. The motivational example presented in the introduction is intuitively clear and make sense, effectively illustrating the core challenge that motivates the overall method.
2. The rationale underpinning CLINIC, particularly its novel compression strategy, is rigorously validated by the theoretical analysis, which provides a solid mathematical foundation for the proposed method.
3. The paper is logically structured and clearly articulated, making it easy to follow. The effectiveness of the proposed CLINIC is substantiated through extensive comparative experiments.
### Weaknesses
1. The intuition behind the assumption in Theorem 3.1, which suggested the left-hand side (which relates to OOD graphs) expected to be smaller than the right-hand side (which relates to ID graphs), should be clearly clarified.
2. The incorporation of multiple components within the proposed framework, including neighborhood retrieval and twin concordance learning, leads to elevated computational demands.
3. The proposed method involves multiple hyperparameters (e.g., the number of nearest neighbors $k_1$, $k_2$, the cluster count $C$, and the loss balancing coefficients $\beta$, $\gamma$). How to select the optimal values of these hyperparameters seems difficult.

---

> ### Author Rebuttal · Authors · 2026-03-30
>
> We are truly grateful for the time you have taken to review our paper, your insightful comments, and support. Your positive feedback is incredibly encouraging for us! In the following response, we would like to address your major concern and provide additional clarification.
>
> > Q1. The intuition behind the assumption in Theorem 3.1.
>
> **R1**: Thank you for your suggestion! The intuition behind the assumption in Theorem 3.1 is rooted in the triangle inequality. Specifically, for OOD samples, we have
> $$
> \mathbb{E}_ {\text{out}} \Big( \| \mu_ {\text{out}} - \mu_ {\phi'(\mathbf{h}_ i)} \|_ 2 \Big)-\mathbb{E}_ {\text{out}} \Big( \| \mu_ {\text{out}} - \mu_ {\phi(\mathbf{h}_ i)} \|_ 2 \Big)
> <
> \mathbb{E}_ {\text{out}} \Big( \| \mu_ {\phi'(\mathbf{h}_ i)} - \mu_ {\phi(\mathbf{h}_ i)} \|_ 2 \Big)
> $$
> where $\phi(\mathbf{h}_i)$ is the cluster assignment of the $i$-th graph and $\phi'(\mathbf{h}_i)$ denotes the nearest cluster center excluding $\phi(\mathbf{h}_i)$. This inequality follows directly from the triangle inequality applied to the distances between cluster centers.
>
> Moreover, since the model is trained on in-distribution (ID) data, it learns well-separated cluster structures for ID samples. As a result, the distance between different cluster centers in the ID setting is expected to be larger than that in the OOD setting, i.e.,
>
> $$
> \mathbb{E}_ {\text{out}} \Big( \| \mu_ {\phi'(\mathbf{h}_ i)} - \mu_ {\phi(\mathbf{h}_ i)} \|_ 2 \Big)
> \leq
> \mathbb{E}_ {\text{in}} \Big( \| \mu_ {\phi'(\mathbf{h}_ i)} - \mu_ {\phi(\mathbf{h}_ i)} \|_ 2 \Big)
> $$
>
> Combining the above two observations, we obtain
>
> $$
> \mathbb{E}_ {\text{out}} \Big( \| \mu_ {\text{out}} - \mu_ {\phi'(\mathbf{h}_ i)} \|_ 2 \Big)-\mathbb{E}_ {\text{out}} \Big( \| \mu_ {\text{out}} - \mu_ {\phi(\mathbf{h}_ i)} \|_ 2 \Big)
> <
> \mathbb{E}_ {\text{in}} \Big( \| \mu_ {\phi'(\mathbf{h}_ i)} - \mu_ {\phi(\mathbf{h}_ i)} \|_ 2 \Big),
> $$
> which is exactly the assumption stated in Theorem 3.1.
>
> > Q2. Complexity analysis.
>
> **R2:** Thanks for the comment! Given the batch size $|\mathcal{B}|$ and a samll number of retrieved neighbors $k_1$, $k_2$ and $k$ ($k_1, k_2, k \ll B$), the overall complexity is linearly related to the product of average graph size and $|\mathcal{B}|^2$, maintaining the same level of computational complexity as GOOD. Furthermore, we have provided the training time, GPU memory, and parameter count of our CLINIC to further demonstrate its computational efficiency.
> | Method | BZR+COX2 (s) | BBBP+BACE (s) | AIDS+DHFR (s) | GPU Memory (M) | Parameter Count (M)|
> | :- | :-: | :-: | :-: | :-: | :-: |
> |GOOD|53|407|424|591.83|0.08|
> |HGOE|57| 429 | 445 | 590.74 |0.07|
> | Ours|54|414|430|602.48|0.10|
>
> > Q3. How to select the optimal values of these hyperparameters seems difficult.
>
> **R3：** Thanks for the comment! We perform a grid search and provide recommended hyperparameters in Parameter Analysis (Section 4.4). Specifically, we search for the number of clusters within {2, 3, 5, 10, 15, 20} and observe that the optimal value depends on the dataset scale. Similarly, we explore the number of k-nearest neighbors within {2, 3, 5, 10, 15, 20} and find that the optimal range is typically {3, 5}. Our results indicate that the model is not highly sensitive to these hyperparameters. Additionally, we set the weights for consistency loss and boundary compression loss to 1. The hyperparameter settings on each dataset are listed in Table 3 in the Appendix.
>
> > Q4. How would the model perform if the method replaced the contextually affinitive neighbors used in the loss calculation with neighbors retrieved from the original Euclidean space?
>
> **R4:** Thanks for your question. We have further compared with a model **variant**, which replaces the neighbors in the loss calculation with those in the original Euclidean space. The experimental results are presented in the following table, and we can find that our model achieves superior performance compared with the variant.
>
> | Method | BZR+COX2 | PTC-MR+MUTAG | AIDS+DHFR | BBBP+BACE | Tox21+SIDER
> | :- | :-: | :-: | :-: | :-: | :-: |
> | variant | 95.17 | 80.73 | 98.41 | 82.49 | 70.02 |
> | Ours | 96.32 | 82.60 | 99.42 | 84.21 | 70.69 |
>
> > Q5. The proposed framework integrates three distinct modules. Among these, which one contributes most significantly to the observed performance gains?
>
> **R5:** Thanks for your valuable comment! Please refer to R1 in Reviewer 7Zrh.
>
> > Q6. The second-best result is marked with an underline in Table 1, but there is a duplication of the underline.
>
> **R6:** We sincerely thank the reviewer for pointing out this typo. We have corrected the duplicated underlines in Table 1 in the revised manuscript.
>
> Thanks again for appreciating our work and for your constructive suggestions. We will properly include all the rebuttal contents in the revised version, following your valuable suggestions.

---

> > ### Author Rebuttal · Reviewer_Ldrb · 2026-04-02
> >
> > Thank you for your detailed reply. I will keep my score.

---

> > > ### Author Response · Authors · 2026-04-02
> > >
> > > Thanks for your positive feedback! We will properly include all the rebuttal contents in the revised version, following your valuable suggestions.

---

### Decision · Program_Chairs · 2026-04-30

**Decision:**

Accept (regular)

**Comment:**

The reviewers are broadly consistent in supporting acceptance. Overall, the paper is viewed as well motivated, technically sound, and empirically strong. Further, the rebuttal has resolved most of the main concerns. Meanwhile, there remain some weaknesses that keep my assessment at the weak-accept level, most notably that the joint online learning process and the construction of a meaningful shared latent space could still be explained in greater detail, both conceptually and technically. Clarifying this part more fully in the final version would further strengthen the paper. So I support a weak accept.